# CoNNect: Connectivity-Based Regularization for Structural Pruning of Neural Networks

**Christian Franssen**[*]                                                            *c.p.c.franssen@vu.nl*
*Vrije Universiteit Amsterdam*

**Jinyang Jiang**[*]                                                            *jinyang.jiang@stu.pku.edu.cn*
*Peking University*

**Yijie Peng**                                                            *pengyijie@pku.edu.cn*
*Peking University*

**Bernd Heidergott**                                                            *b.f.heidergott@vu.nl*
*Vrije Universiteit Amsterdam*

**Reviewed on OpenReview:** *https://openreview.net/forum?id=RIZCe7BuEp*

## Abstract

Pruning encompasses a range of techniques aimed at increasing the sparsity of neural networks (NNs). These techniques can generally be framed as minimizing a loss function subject to an $L_0$ norm constraint. This paper introduces CoNNect, a novel differentiable regularizer for sparse NN training that ensures connectivity between input and output layers. We prove that CoNNect approximates $L_0$ regularization, while preserving essential network structure and preventing the emergence of fragmented or poorly connected subnetworks. Moreover, CoNNect is easily integrated with established structural pruning strategies. Numerical experiments demonstrate that CoNNect can improve classical pruning strategies and enhance state-of-the-art one-shot pruners, such as DepGraph and LLM-pruner.

## 1 Introduction

Machine learning models, such as neural networks (NNs), have seen rapid growth in recent years, leading to significant increases in model performance. However, as the footprint of these models increases (Patterson et al., 2021), there is a growing need to develop more energy-efficient approaches to machine learning that can balance computational performance with environmental sustainability.

An effective technique for reducing a model's computational effort and memory burden is *neural network pruning*. Pruning refers to the process of systematically eliminating parameters that contribute little to network performance. The resulting sparse NNs have attracted significant interest in recent years due to their ability to boost computational efficiency and minimize memory consumption while preserving or even improving model performance (LeCun et al., 1989; Hassibi et al., 1993; Frankle & Carbin, 2019).

Various techniques have been proposed to achieve sparsity in NNs, such as unstructured pruning, which involves selectively removing individual weights from the network. Pruning weights based on their individual magnitudes is a classic example of unstructured pruning (LeCun et al., 1989; Hassibi et al., 1993; Hagiwara, 1993; Han et al., 2015), where connections are removed solely based on the absolute value of each weight. Although it can provide highly sparse networks, it often results in irregular memory access patterns, which can be difficult to optimize in hardware implementations. Recently, semi-structured pruning has emerged as an approach to balance granularity and efficiency by removing weights within predefined patterns or

---

[*]Corresponding authors.

groups (Frantar & Alistarh, 2023; Sun et al., 2024; Fang et al., 2024). While this is a promising approach for realizing high accuracy at various sparsity ratios, it presents nuanced trade-offs in inference speed and therefore the computational efficiency of the model. Finally, structured pruning (e.g., see Yuan & Lin (2006); Huang & Wang (2018); Anwar et al. (2017)) offers a systematic method to remove entire groups or channels of neurons. Techniques like Group Lasso (Yuan & Lin, 2006; Hoefler et al., 2021) and other structured sparsity learning (Wen et al., 2016; Zhuang et al., 2020) fall into this category; see He & Xiao (2023) for a review. The structured removal of parameters generally leads to an almost equal reduction in computational complexity and inference speed, thus immediately improving computational efficiency. For instance, both at a 50% pruning rate, structured pruning (Ma et al., 2023) achieves a 1.85× end-to-end latency acceleration on LLaMA-7B (Touvron et al., 2023), whereas semi-structured pruning (Sun et al., 2024) only achieves a 1.24× speedup. Structured pruning is the only paradigm that enables a universal neural network compression without requiring special hardware or software, thereby addressing the primary objective of pruning: acceleration (Cheng et al., 2024).

This highlights the practical value of structured pruning, motivating a deeper investigation into the principles that should guide pruning strategies. We believe that pruning should obey the following two axioms (where we identify a NN with a directed, weighted graph):

**Axiom 1** (Delete Weights to Improve Computational Efficiency). *The graph should be 'small': pruning must significantly reduce the number of weights while minimally impacting accuracy and maximizing computational efficiency.*

**Axiom 2** (Preserve Neural Network Connectivity). *The pruning process must prevent disruptions in the connectivity of the neural network and preserve the flow of information from input to output.*

The extensive research on pruning neural networks, as more elaborately outlined in the literature overview in Section 2 and particularly in review works such as Hoefler et al. (2021); He & Xiao (2023), predominantly aligns with the first axiom. However, few methods address Axiom 2, as the impact of weight removal on overall network connectivity is rarely considered. This negligence can result in a pruning that produces highly disconnected networks (Vysogorets & Kempe, 2023), or in the most extreme case so-called *layer collapse*, see Figure 1, where the NN becomes completely dysfunctional. A notable exception is SynFlow pruning (Tanaka et al., 2020), a method designed for unstructured pruning at initialization that explicitly considers connectivity by preserving signal flow through the network. We explore SynFlow in more detail in Section 3.3.1.

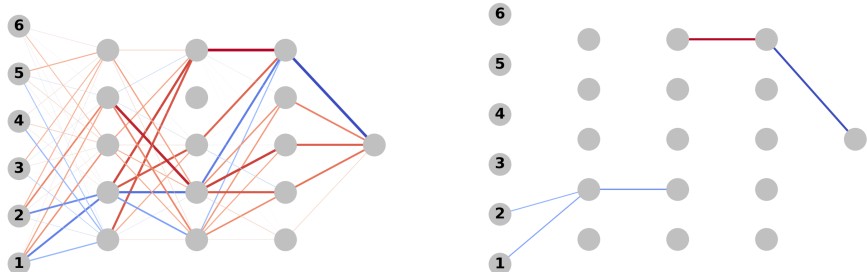

Figure 1: Magnitude-based pruning of NN (left) leads to layer collapse (right).

In this paper, we propose a new regularizer, called CoNNect, that can be used to satisfy both axioms simultaneously and (i) is differentiable (except in the point zero) and allows for gradient descent optimization, (ii) effectively approximates $L_0$ regularization and guarantees maximally connected network structures as stable stationary points, avoiding issues such as layer collapse. CoNNect is best understood as a generalization of the SynFlow principle into a training-time regularizer that promotes network connectivity throughout optimization. It utilizes a weight normalization for its measurement, resulting in weights being restricted to $[0, 1]$, so that the contribution of a path from the input to the output layer to the overall connectivity of the network goes exponentially quickly to zero unless the weights along the paths are (close to) 1. Hence, when maximizing the connectivity for the normalized weights, we find a weight association that prefers fewer

*direct paths* over many *parallel paths*, while focusing on the connectivity of the input with the output layer. Importantly, we show how CoNNect is applicable for structural pruning, and seamlessly integrates within one-shot pruning pipelines.

We demonstrate CoNNect's efficacy through a series of numerical examples. First, we provide an illustrative unstructured pruning example in which we show that pruning strategies like magnitude-pruning (LeCun et al., 1989; Hassibi et al., 1993) and SynFlow (Tanaka et al., 2020) can benefit from CoNNect regularization during training. Here, it outperforms $L_1$ and $L_2$ regularization in terms of both accuracy and stability (measured by the frequency of layer collapse). Then, we specifically conduct numerical experiments on the integration of CoNNect within structured pruning, given its potential to achieve substantial improvements in computational efficiency, thus satisfying Axiom 1. We apply CoNNect regularization on a channel level when training a Graph Neural Network (GNN), achieving improved performance compared to $L_1$ and $L_2$ regularization. Then, we integrate CoNNect into state-of-the-art methods for structural pruning of pre-trained models, such as DepGraph (Fang et al., 2023), and LLM-pruner (Ma et al., 2023), a one-shot pruning method for Large Language Models (LLMs). Our numerical results demonstrate consistent performance improvements compared with other methods obtaining similar gains in computational efficiency.

## 2 Related Work

The concept of pruning NNs dates back to the early 1990s. The seminal work by LeCun et al. (1989) on Optimal Brain Damage introduced the idea of pruning by removing weights that contribute least to performance, thus simplifying the network. Hassibi et al. (1993) extended this concept with Optimal Brain Surgeon, which provided a more sophisticated method for determining which weights to prune based on their impact on the error function. These early methods laid the foundation for modern pruning techniques, focusing on reducing network complexity while maintaining accuracy.

**Regularization-Based Pruning (Soft Pruning)**. Regularization methods play a crucial role in promoting sparsity by extending the loss function with a penalty function that discourages overly complex models. These methods typically either encourage certain components (e.g., weights or filters) to become ineffective, thereby yielding sparsity, or drive them to become functionally redundant, enabling safe removal after training (Ding et al., 2019; Valverde et al., 2024). In the case where regularization encourages sparsity, regularization does not explicitly set the weights to zero but instead reduces their magnitude, allowing them to remain non-zero and potentially become active again if needed. This leads to what is termed soft pruning, where sparsity is encouraged but not strictly enforced through hard weight removal during training. After training concludes, unimportant weights, typically those with the smallest magnitudes, are then pruned (Hagiwara, 1993; Gale et al., 2019). One of the simplest and most widely used methods, $L_1$ regularization (Tibshirani, 1996; He et al., 2017; Yang et al., 2019; De & Doostan, 2022; Ziyin & Wang, 2023), penalizes the sum of the absolute values of the weights, encouraging many weights to become zero. Moreover, $L_1$ regularization fails to incorporate considerations from Axiom 2, which emphasizes the preservation of neural network connectivity and functionality. This lack of consideration for connectivity can lead to a network that, while sparse, may suffer from disrupted information flow, ultimately impairing its performance. Similarly, $L_2$ regularization, another common regularization technique, penalizes the sum of the squares of the weights (e.g., see Hinton (2012); Phaisangittisagul (2016); Loshchilov et al. (2017)). While $L_2$ regularization is effective at discouraging large weights, it does not push small weights towards zero, thus failing to induce sparsity in the network. As a result, $L_2$ regularization typically produces networks with small but non-zero weights, which do not benefit from the same computational efficiency gains that a sparse network would offer. Moreover, like $L_1$ regularization, $L_2$ regularization does not address the need to maintain critical connections as highlighted by Axiom 2, making it less suitable for tasks where maintaining network connectivity is essential.

**Stage-Based Pruning (Hard Pruning)**. Stage-based pruning strategies are utilized as separate, discrete actions during various stages of model training. These techniques can be implemented before training (Lee et al., 2019; Tanaka et al., 2020; Wang et al., 2020), during training (Frankle & Carbin, 2019; Mocanu et al., 2018; Jayakumar et al., 2020), or after training (Hagiwara, 1993; Thimm & Fiesler, 1995; Gale et al., 2019; He et al., 2019; Ma et al., 2023). Stage-based pruning generally does not fundamentally alter

the objective function or the descent direction like regularization does, but instead acts on the model's structure or parameters at specific moments. These kinds of pruning methods can be considered hard pruning approaches, as parameters are explicitly removed. Many different criteria for pruning have been introduced, such as magnitude-based pruning (Hagiwara, 1993; Gale et al., 2019), which involves removing weights with the lowest absolute values and is based on the idea that these weights have the least impact on the overall performance of the model. More complex criteria have been constructed to determine the impact of weight removal, such as first-order (e.g., see Zhou & Si (1999); Molchanov et al. (2017); Sanh et al. (2020)) and second-order expansions (LeCun et al., 1989; Hassibi et al., 1993; Ma et al., 2023) of the training objective. Specifically, SynFlow (Tanaka et al., 2020) is a method that adheres closely to the principles of Axiom 2, focusing on retaining the network's connectivity and functionality during pruning. Unlike magnitude-based techniques, SynFlow utilizes a first-order expansion of signal flow to pinpoint and remove weights with minimal impact on the network's overall information flow. This approach ensures that while the network is being pruned, its structural integrity is preserved and the critical pathways in terms of connectivity remain intact. Another approach adopting a network-theoretic perspective is Li et al. (2020), who employ Katz centrality to prune neural network nodes in nonlinear system modeling. Although this method highlights the potential of network measures for guiding pruning decisions, our methodology is fundamentally different and further extends to large-scale NNs.

We conclude the above discussion by noting that the CoNNect regularizer, to be introduced in the next section, can both be used as a soft pruning approach and integrated in hard pruning approaches.

## 3 Methodology

### 3.1 Preliminaries

We define a graph $\mathcal{G} = (V, E)$, where $V$ denotes the set of vertices (or nodes) and $E$ represents the set of directed links that connect these vertices. A weighted graph has weights $W_{i,j} \geq 0$ for links $(i, j) \in E$, where we let $W_{i,j} = 0$, for $(i, j) \notin E$. Neural networks can be described using graph theory by representing them as directed, weighted graphs. In this setting, the vertices $V = V_1 \cup \ldots \cup V_K$ in the graph correspond to the neurons in the network which are organized into distinct subsets corresponding to the different layers $V_k$, for $k = 1, \ldots, K$. Here, the input nodes $V_1$ represent the neurons in the input layer, the hidden nodes $V_k$, for $k = 2, \ldots, K-1$, represent the neurons in the hidden layers, and the output nodes $V_K$ represent the neurons in the output layer.

Throughout the paper, we describe a neural network $\mathcal{G}$ using the tuple $(W, b)$, where $W \in \mathbb{R}^{|V| \times |V|}$ is the weighted adjacency matrix of the weights, such that $W_{i,j}$ connects node $i \in V_k$ with node $j \in V_{k+1}$, and $b = (b_1, \ldots, b_{|V|})$ is the bias vector. Moreover, we denote the activation of the $k + 1$th layer by the tensor $X^{(k+1)} = \sigma(W^{(k)} X^{(k)} + b^{(k+1)})$, where $\sigma$ is the activation function, $W^{(k)}$ is the submatrix containing the weights between nodes in $V_k$, and $V_{k+1}$, and $b^{(k+1)}$ the biases for the nodes in $V_{k+1}$. Finally, we denote $f(X^{(1)}; W, b)$ as a forward pass through the network. Note that, for notational simplicity, we omit untrainable structures in neural networks, such as residual connections, which will be dealt with in later sections. However, they can be incorporated by treating them as edges with fixed weights.

### 3.2 Problem Formulation

Let $\{(x_i, y_i)\}_{i=1}^N$ denote the training set, where $x_i = X_i^{(1)}$ represents the input data and $y_i$ represents the corresponding label for each of the $N$ samples. Fitting the parameters of a neural network $\mathcal{G}$ involves optimizing the network's weights to minimize a loss function $\mathcal{L}(\hat{y}, y)$, where $\hat{y} = f(x; W, b)$ is the predicted output given an input $x$.

In this paper, our objective is to train a sparse neural network, which can be achieved by inducing sparsity in the network's parameters. A commonly employed approach to sparsification is regularization. Regularization involves augmenting the loss function with an additional term that penalizes non-zero elements in the network

parameters. Specifically, the optimization problem can be formulated as:

$$\min_{W,b} \quad \mathcal{L}(\hat{y}, y) + \lambda R(W),$$ (1)

where $R(W) = \|W\|_{0,1}$. However, this $L_0$ norm is non-convex and leads to a combinatorial optimization problem, which is generally NP-hard and computationally intractable for large-scale problems. A more practical alternative is $L_1$ regularization, as in Lasso regression, where $R(W) = \|W\|_{1,1}$. $L_1$ regularization induces sparsity by shrinking weights to zero, approximating the $L_0$ norm while remaining convex and suitable for gradient-based optimization. However, $L_1$ regularization primarily satisfies Axiom 1 by reducing connections but fails to address Axiom 2, which focuses on preserving network connectivity and ensuring efficient signal flow. This limitation can result in a disconnected or underperforming network when key pathways are not maintained.

### 3.3 CoNNect

To overcome the aforementioned issues, we propose CoNNect, a regularizer that considers the network's overall connectivity, ensuring that the structure contributes to optimal performance. While we first define CoNNect in an unstructured form to establish it from the ground up in Section 3.3.1, this formulation serves as a foundation: scaling factors can be introduced to control the granularity of pruning and seamlessly transition to structured regularization, as we detail in Section 3.3.2. This design enables CoNNect to directly support structured pruning objectives while maintaining flexibility and precision.

### 3.3.1 Weight-Level Regularization

Katz centrality is a measure used in network analysis to determine the relative connectivity of a node in a network by considering both the number and the quality of connections (Katz, 1953). Inspired by the connectivity measurement in Katz centrality, let us consider the following connectivity matrix for a neural network:

$$\varphi(W) = \sum_{k=1}^{K} (\theta(W))^k,$$

where $(\varphi(W))_{i,j}$ indicates the connectivity from node $i$ to node $j$. The matrix $\theta(W)$ is a normalized representation of the network's parameterized weights between successive layers, capturing the relative strength of connections. For $i \in V_k$ and $j \in V_{k+1}$, the normalized weight is defined as

$$(\theta(W))_{i,j} = \frac{|W_{i,j}|}{\sum_{(k,l) \in E_k} |W_{k,l}|},$$ (2)

where $E_k$ denotes the set of edges connecting the $k$th layer $V_k$ to layer $V_{k+1}$, and $W_{i,j}$ is the parameterized weight of the connection between nodes $i$ and $j$. Then, in the context of a neural network, we can denote the connectivity by taking the sum of connectivity values between the input and output layers:

$$\varphi^{tot}(W) = \sum_{i \in V_1} \sum_{j \in V_K} (\varphi(W))_{i,j}.$$

Finally, we argue for the preservation of connectivity (as per Axiom 2), so we aim to maximize the network's overall connectivity. Consequently, we choose the regularizer as:

$$R(W) = -\varphi^{tot}(W),$$ (3)

which we will refer to as the CoNNect regularizer.

It is important to note that Equation (3) exclusively considers parameterized weights of the network. Since structures like residual connections, which are commonly used in modern neural network architectures, do not contain learnable parameters, they are therefore excluded. This ensures that the CoNNect regularizer exclusively influences the learnable parameters of the network without conflating it with architectural shortcuts that are not subject to optimization. In practice, this means CoNNect promotes sparse yet effective

parameter usage without penalizing or being affected by static residual pathways, thereby exclusively focusing the regularization on the components that matter for model capacity and generalization. We refer the reader to Appendix B for additional implementation details, including the treatment of activation functions, pooling layers, and other architectural components.

CoNNect is effectively the (negative of the) sum of all (multiplicative) normalized weighted paths between nodes in the input layer $V_1$ and the output layer $V_K$. It follows that $-\varphi^{tot}(W) = 0$ if and only if there is no path with positive parameterized weights between the input and output layer. Moreover, $-\varphi^{tot}(W)$ can be efficiently computed using a single forward pass $f(\bar{1}, W, \bar{0})$, where $\bar{1}$ is a vector of ones as input, $\bar{0}$ is a vector of zeroes for the biases, and finally taking the sum of the output values. Thus, for a batch size of $M$, the additional time used for computing $\varphi^{tot}(W)$ is proportional to $\frac{1}{M}$. Hence, CoNNect can be efficiently applied to large-scale neural networks without incurring significant computational overhead.

In the following, we show that $-\varphi^{tot}(W)$ can be used as a surrogate regularizer for the $L_0$ norm to induce sparsity. Taking $R(W) = ||W||_{0,1}$ in Equation (1), it is easy to show that any neural network $W$ that minimizes $||W||_{0,1}$ while connecting the input layer to the output layer via parameterized weights, i.e., $\varphi^{tot}(W) > 0$, has $K - 1$ non-zero weights. As the following theorem shows, a similar result holds for the CoNNect regularizer as any $W$ minimizing $-\varphi^{tot}(W)$ has only $K - 3$ non-zero weights between layers 2 and $K - 1$.

**Theorem 3.1.** *Consider the problem*

$$\min_W \; -\varphi^{tot}(W), \tag{4}$$

*for a given network with number of layers $K > 2$. All solutions $W^*$ to Equation (4) have at most $|V_1| + |V_K| + K - 3$ non-zero weights.*

*Proof.* See Appendix A.1. □

Theorem 3.1 demonstrates that $L_0$ norm regularization can be effectively achieved through the CoNNect regularizer, as the induced sparsity in large neural networks is comparable. Importantly, the difference in the number of non-zero elements becomes negligible in practice when most input nodes contribute valuable predictive information, and all output nodes are used for accurate classification. Also, the regularizer does not cause input nodes to disconnect, as it does not depend on how many input nodes are connected to the second layer. This is a beneficial feature. If certain input nodes were disconnected, as might happen with other regularizers such as $L_1$ regularization, important data features could be disregarded, potentially resulting in suboptimal model performance.

We now show that CoNNect is a well-behaved regularizer in the sense that it does not have stable stationary points other than its global optima. This ensures that if a gradient descent gets stuck in a stationary point of the regularizer, the loss function will always push the solution to leave the stationary point unless the loss function itself is stationary at that point. In the following, we exclusively consider connected $W$, that is, $\varphi^{tot}(W) > 0$. We do so because with the log transformation the objective diverges as connectivity vanishes, preventing disconnection during training.

First, consider for some $(i, j) \in E_k$ let

$$\partial_{W_{i,j}}(\theta(W))_{i,j} = \frac{\sum_{(r,c) \in E_k} |W_{r,c}| - |W_{i,j}|}{(\sum_{(r,c) \in E_k} |W_{r,c}|)^2},$$

and, specifically for $(q, t) \neq (i, j) \in E_k$;

$$\partial_{W_{q,t}}(\theta(W))_{i,j} = \frac{-|W_{i,j}|}{(\sum_{(r,c) \in E_k} |W_{r,c}|)^2}.$$

Observe that differentiating $\theta(W)$ with respect to $W_{i,j}$ only affects the weights in the same layer as $W_{i,j}$. Thus, a stationary point to Equation (4) solves the following first-order conditions:

$$\sum_{(r,c)\in E_1} \partial_{W_{i,j}}(\theta(W))_{r,c} \cdot a_{c\cdot} = 0, \quad \forall \, (i,j) \in E_1, \tag{5}$$

$$\sum_{(r,c)\in E_2} a_{\cdot r} \cdot \partial_{W_{i,j}}(\theta(W))_{r,c} \cdot a_{c\cdot} = 0, \quad \forall \, (i,j) \in E_2,$$

$$\vdots$$

$$\sum_{(r,c)\in E_{K-1}} a_{\cdot r} \cdot \partial_{W_{i,j}}(\theta(W))_{r,c} = 0, \quad \forall \, (i,j) \in E_{K-1},$$

where

$$a_{\cdot r} = \sum_{i\in V_1} \sum_{\gamma\in\Gamma_{i,r}} \prod_{k=1}^{|\gamma|-1} (\theta(W))_{\gamma_k}, \quad a_{c\cdot} = \sum_{m\in V_K} \sum_{\gamma\in\Gamma_{c,m}} \prod_{k=1}^{|\gamma|-1} (\theta(W))_{\gamma_k},$$

are the connectivity from input layer to a node $r$ and connectivity from a node $c$ to the output layer, respectively. To satisfy Equation (5), we need:

- the weights for the edges in $E_1$ must be assigned to all $(\theta(W))_{i,j}$, where $j \in \arg\max_p a_p$;

- the weights for the edges in $E_k$, $k = 2, \ldots, K-2$ must be assigned to $(\theta(W))_{i,j}$, where $(i,j) \in \arg\max_{(p,q)} a_{\cdot p} a_{q\cdot}$;

- the weights for the edges in $E_{K-1}$ must be assigned to $(\theta(W))_{i,j}$, where $i \in \arg\max_q a_{\cdot q}$.

The set of weight matrices $W$ that satisfy Equation (5) can be more precisely formulated as in Lemma 3.2.

**Lemma 3.2.** *Assume a neural network with $K > 3$ layers. All stationary points $W^*$ to Equation (4) that are connected, i.e., $\varphi^{tot}(W) > 0$, have paths with equal subsequent weights between layers $2$ and $K-1$ on its non-zero paths. That is, for each two paths $\gamma', \gamma'' \in \bigcup_{i\in V_1, m\in V_K} \Gamma_{i,m}$, such that*

$$\prod_{k=1}^{K-1} (\theta(W^*))_{\gamma_k} > 0, \quad \gamma \in \{\gamma', \gamma''\},$$

*i.e., both paths have positive weight, we have $(\theta(W^*))_{\gamma'_k} = (\theta(W^*))_{\gamma''_k}$, for all $k = 2, \ldots, K-2$.*

*Proof.* See Appendix A.2. $\qquad\square$

Using Lemma 3.2, we note that all non-optimal stationary points, i.e., $\varphi^{tot}(W) < 1$, have multiple directions of improvement by simply transferring mass from one path to another. It follows that these solutions are inherently unstable and thus are not a local optimum. We present a precise statement in Theorem 3.3, where we the proof is omitted as it follows directly form the previous observation.

**Theorem 3.3.** *Assume a neural network with $K > 3$ layers. All stable stationary points $W^*$ to Equation (4) that are connected, i.e., $\varphi^{tot}(W) > 0$, are global minimizers.*

As Theorem 3.3 shows, the only stable stationary points of CoNNect are those where the weight matrix has only $K - 3$ non-zero weights between layer $2$ and $K - 1$. Moreover, CoNNect does not have regions of attraction, and thus is a well-behaved regularizer for gradient search.

As argued earlier, it is recommended to take the logarithm of the connectivity regularizer, i.e.,

$$-\log\left(\varphi^{tot}(W)\right), \tag{6}$$

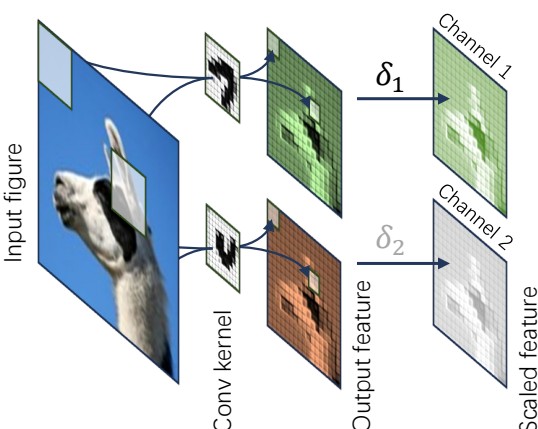

Figure 2: Illustration of CNN with the scaling factor.

as it ensures that if the neural network tends to disconnect during training, i.e., $\varphi^{tot}(W) \to 0$, Equation (6) approaches $\infty$, hence preventing layer collapse. Moreover, it enhances numerical stability, ensuring that the regularization term remains well-behaved even for varying scales of connectivity.

Once we have trained a model with CoNNect regularization, many of the redundant weights will have been pushed to zero. Consequently, we can hard prune the regularized model using pre-established pruning strategies. A well-known strategy is simple magnitude-based pruning (LeCun et al., 1989), which prunes the smallest weights in absolute value. Alternatively, we can use SynFlow pruning (Tanaka et al., 2020), which prunes the neural network's weights according to synaptic saliency scores:

$$I_{i,j} = \left(\partial_{(\theta(W))_{i,j}} \varphi^{tot}(W)\right) \cdot (\theta(W))_{i,j} = a_{\cdot i} \cdot (\theta(W))_{i,j} \cdot a_{j \cdot},$$

and eliminate the weights with the smallest $I_{i,j}$ values.

### 3.3.2 Channel-Level Regularization

The regularizer introduced in Section 3.3.1 was explicitly defined on the weights of the neural network (effectively making it an unstructured pruning approach). In this section, we show the required modifications to extend it to structured pruning. To this end, we can introduce a scaling factor for the output of structures (e.g., neurons, channels, etc.) that we want to prune (Huang & Wang, 2018). In the following, we explain how to include structured pruning at the channel level in, e.g., Convolutional Neural Networks (CNNs) and Graph Neural Networks (GNNs), but this can be naturally extended to any parallel structures in neural networks, such as nodes, but also entire block structures.

Neural networks that utilize channels are designed to process multi-dimensional data where information is structured across multiple feature dimensions, such as color channels in images or frequency bands in audio. For example, CNNs are a specialized type of neural network designed to process grid-like data such as images. These images can be represented using a tensor $X \in \mathbb{R}^{d \times h \times w}$, where $d$ refers to the number of channels (e.g., RGB for color images) and $h$ and $w$ refer to the height and width of the image respectively. A standard CNN consists of (several) convolutional layers followed by an activation function (e.g., ReLU), and pooling layers that reduce spatial dimensions while preserving important features. Convolutional layers transform the tensor into a set of feature maps through a series of learned filters (also known as kernels). Each convolutional layer in the CNN applies these filters to local regions of the input, capturing spatial hierarchies and patterns like edges, textures, and more complex shapes as the network deepens.

For performing regularizing at the channel level, we introduce a set of learnable parameters that scale the output of each channel after a convolutional layer. More formally, for every $X^{(k)} \in \mathbb{R}^{d \times h \times w}$, which is the activation after the $k$-th convolutional layer, we scale the channels with $\delta^{(k)} \in \mathbb{R}^d$ so that $X^{(k)\prime} = \delta^{(k)} \odot X^{(k)}$, where $\odot$ denotes element-wise multiplication so that the scaling factor $\delta^{(k)}$ is broadcast across the height $h$ and width $w$. The inclusion of scaling factors $\delta^{(k)}$ is a simple linear transformation and so can be perceived as the introduction of an additional layer to the neural network $W$, see Figure 2, resulting in an extended

neural network denoted by $W'$. As the normalization in Equation (2) will also be applied on the scaling factors, the unstructured CoNNect regularizer in Equation (3) carries over to a structured regularization, where the scaling factors of less informative channels are pushed to 0 and more informative channels are retained.

Once a regularized neural network is obtained, we can do pruning in a similar fashion as in Section 3.3.1. Specifically, we can prune its channels via calculating an importance scores for each channel. To that end, we aim to determine the contribution of a channel $c$ in layer $k$ in terms of the connectivity of the neural network, denoted by $I_{k,c}$. More formally, let $\theta_c^{(k)}(\delta) = |\delta_c^{(k)}|/\|\delta^{(k)}\|_1$ denote the normalization of the scaling factors with index $c$ for convolutional layer $k-1$ so that $I_{k,c}$ can be determined via

$$I_{k,c} = \left(\partial_{\theta_c^{(k)}(\delta)} \varphi^{tot}(W)\right) \cdot \theta_c^{(k)}(\delta) = \left(\sum_{r \in V_{k-1}^{(c)}} a_{\cdot r}\right) \cdot \theta_c^{(k)}(\delta) \cdot \left(\sum_{r \in V_{k+1}^{(c)}} a_{r \cdot}\right),$$

where $V_k^{(c)}$ is the subset of nodes in a layer $k$ corresponding to channel index $c$. Simply put, $I_{k,c}$ denotes the total connectivity that flows through channel $c$ in layer $k$. Consequently, a simple pruning strategy is to prune the channels with lowest values of $I_{k,c}$.

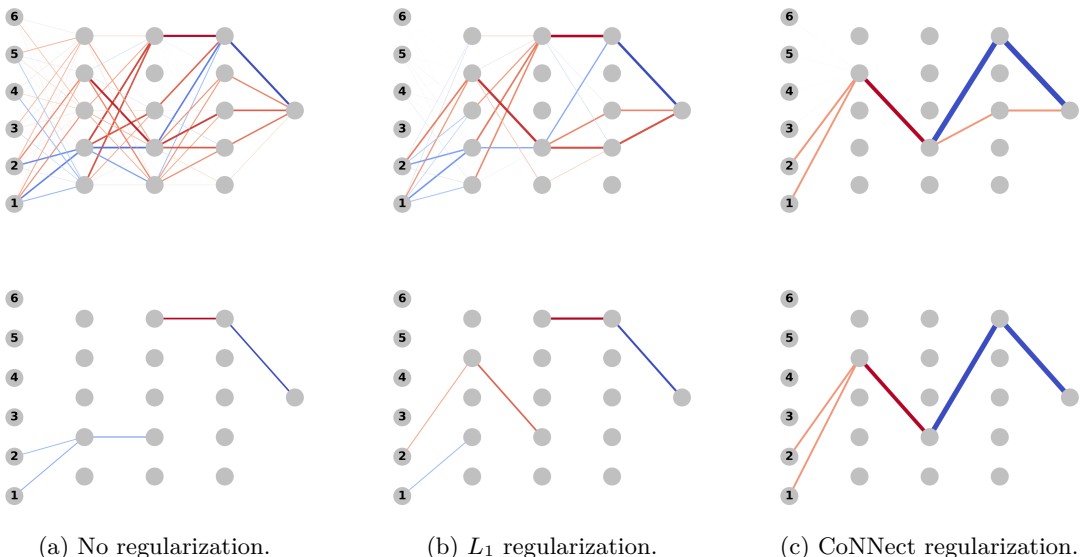

(a) No regularization.    (b) $L_1$ regularization.    (c) CoNNect regularization.

Figure 3: Trained (top) and fine-tuned (bottom) models. Thicker and darker colors correspond to stronger values. Red and blue edges correspond to positive and negative values respectively.

# 4    Numerical Experiments

We now provide several numerical experiments. In Section 4.1, we show results for CoNNect regularization during training. In Section 4.2, we show how CoNNect can be further scaled for pruning pre-trained models through an integration in hard pruning strategies, such as DepGraph (Fang et al., 2023) and LLM-pruner (Ma et al., 2023). Our code is available at `https://github.com/cfn420/CoNNect`.

## 4.1    Regularized Training with CoNNect

### 4.1.1    Unstructured Pruning of MLPs

In the following, we want to study the effects of integrating CoNNect regularization in an unstructured pruning task. Let us consider a small multilayer perceptron neural network with ReLU activations. The network has 6 input nodes, three hidden layers of five nodes, and a single output node. We sample input values $x_i = (x_{i,1}, \ldots, x_{i,6}) \sim \mathcal{N}(0, \Sigma)$, where $\Sigma$ is a matrix with the value 2 on the diagonal. Furthermore,

we let the output values be $y_i = 1$ if $x_{i,1} + x_{i,2} + \xi_i > 0$, and $y_i = 0$ otherwise, where $\xi_i \sim \mathcal{N}(0, 0.25)$. To find a sparse network representation, we train the network with $L_1$ and CoNNect regularization. To that end, we solve

$$\min_{W,b} \ \mathcal{L}(\hat{y}, y) + \lambda_1 \|W\|_{1,1} - \lambda_2 \log\left(\varphi^{tot}(W)\right) + \lambda_3 \|W\|_{2,1}. \tag{7}$$

We fit three different models for 200 epochs following Equation (7), for which we provide coefficients in Table 3, see Appendix C.1. We show the resulting NNs on the top row in Figure 3 for a single neural network initialization. In the bottom row, we present the fine-tuned NNs after SynFlow pruning. As can be seen, the CoNNect regularizer is capable of identifying the relevant paths, where the other methods fail. We provide more details of this experiment in Appendix C.1, including more extensive results.

### 4.1.2 Channel-Level Pruning on GNNs

In this section, we demonstrate CoNNect for structured pruning on the channel level. Specifically, we prune a Graph Convolutional Network (GCN, Kipf & Welling, 2017) containing seven layers with learnable parameters, where the hidden feature dimensions are 512-256-256-256-256-64. Each GCN layer is followed by a ReLU activation function. We train the model on the Cora dataset (Sen et al., 2008), a graph-based dataset consisting of 2,708 academic papers (nodes) and 5,429 citation links (edges), with each paper categorized into one of seven topics and represented by a 1,433-dimensional binary feature vector.

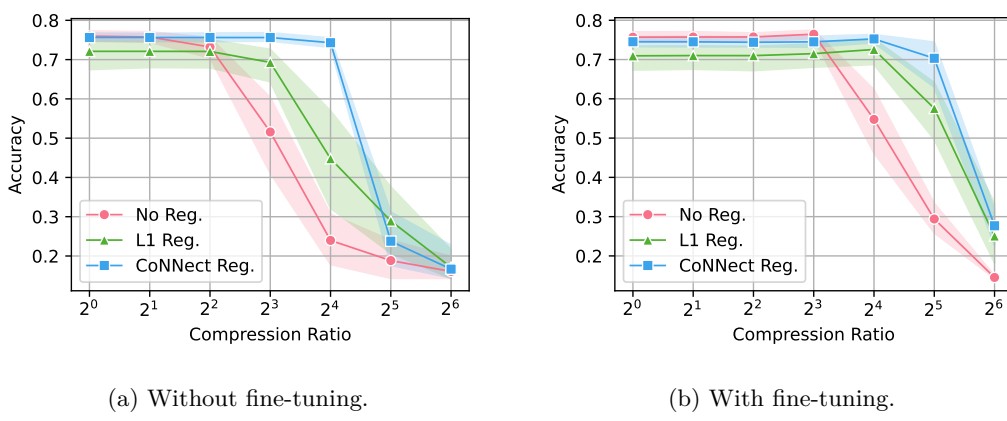

(a) Without fine-tuning.      (b) With fine-tuning.

Figure 4: Accuracies of GNNs for given pruning ratios.

We train GCNs following Equation (7) for 300 epochs using the parameters shown in Table 4 in Appendix C.2 and fine-tune each model after pruning for 100 epochs. We conduct 10 repeated experiments, and as shown in Figure 4, our method outperforms $L_1$ regularization, especially for high compression ratios, which are computed by total channels/(total channels - pruned channels). The shaded regions represent 98% confidence intervals. We refer the reader to Appendix D.2 for an extensive ablation on the regularizer coefficients.

## 4.2 Pre-Trained Model Pruning with CoNNect

To further demonstrate the versatility and scalability of CoNNect, we integrate it into DepGraph (Fang et al., 2023) and LLM-Pruner (Ma et al., 2023). These frameworks ensure all parameters are divided into several groups according to the dependency relationships in the computation process. Then, the importance score under the objective function $\mathcal{J}(\cdot)$ is calculated by $I_{i,j} = |\mathcal{J}_{W_{i,j}}(W) - \mathcal{J}_{W_{i,j}=0}(W)| \approx |\partial_{W_{i,j}} \mathcal{J}(W) \cdot W_{i,j}|$. We integrate our CoNNect approach through the objective, i.e., $\mathcal{J}(W) = \mathcal{L}(\mathcal{D}) - \lambda \log(\varphi^{tot}(W))$, where $\mathcal{D}$ denotes the dataset. The importance of each group is aggregated through summation, and the least important groups are pruned. Connectivity, as currently defined in CoNNect, is by default not affected by modules such as activation functions or biases. However, when keeping such modules in place within the connectivity computation, CoNNect can be understood in terms of signal flow. In the remainder of this section, we simplify the connectivity computation and only redefine $(\theta(W))_{i,j} = |W_{i,j}|$ to enhance both

numerical stability and computational efficiency, while also setting the biases to $|b|$. The result is that we aim to prune the groups that minimally affect the loss function, while also preserving signal flow.

### 4.2.1 One-shot Pruning CNNs

Using our integration in DepGraph, we perform structural pruning on ResNet-56 (He et al., 2016) and VGG-19 (Simonyan & Zisserman, 2015), which are pre-trained and fine-tuned on CIFAR-10 and CIFAR-100 datasets (Krizhevsky, 2009), respectively (see Appendix C.3). DepGraph framework iteratively prunes the model until the predefined speed-up targets are achieved, which is calculated as the ratio of multiply-accumulate operations before and after pruning. We first follow the pruning intensities tested in Fang et al. (2023), and then verify CoNNect further with extreme cases. Thus, the pruning is set to target speed-ups of 2.5× and 16× for ResNet-56 on CIFAR-10 and 8×, and 16× for VGG-19 on CIFAR-100. As shown in Table 1, CoNNect exhibits advantages across various pruning ratios, with benefits being more pronounced in more extreme cases.

Table 1: Pruning results on ResNet-56 and VGG-19.

| MODEL & DATASET | BASE ACC. | METHOD | PRUNED ACC. | SPEED UP | PRUNING RATIOS |
|---|---|---|---|---|---|
| RESNET-56 & CIFAR-10 | 93.53 | DEPGRAPH | 93.17 | 2.51× | 56.22 |
| | | CONNECT | **93.63** | 2.50× | 53.20 |
| | | DEPGRAPH | 80.24 | 16.17× | 98.27 |
| | | CONNECT | **83.12** | 17.24× | 97.46 |
| VGG-19 & CIFAR-100 | 73.50 | DEPGRAPH | 65.89 | 8.12× | 90.48 |
| | | CONNECT | **69.38** | 8.00× | 93.33 |
| | | DEPGRAPH | 57.48 | 16.10× | 96.14 |
| | | CONNECT | **62.56** | 16.07× | 97.51 |

### 4.2.2 One-shot Pruning LLMs

We adopt the structural definition of connectivity as proposed in CoNNect and incorporate it into the LLM-pruner framework (Ma et al., 2023). In our LLM-pruner integration, we compute the connectivity score for each parameter group based on its structural position and its contribution to information propagation within the transformer architecture. This metric is estimated using model parameter dependencies derived from a small calibration set, as described in Appendix C.4. This approach allows us to efficiently integrate CoNNect into large-scale models, capturing essential structural information without modifying the architecture or relying on activation-based analysis.

Now, we perform a one-shot pruning on LLaMA-7B (Touvron et al., 2023) using our CoNNect integration in LLM-pruner. After pruning, the LLM is fine-tuned with LoRA (Hu et al., 2022) to restore as much structural capability as possible under the current architecture. To assess the model performance, we conduct a zero-shot perplexity analysis, see Table 2. We first compare CoNNect to $L_1$, $L_2$, random, and vanilla LLM-Pruner's importance metrics with 25% of the parameter groups removed, thereby resulting in a 20% parameter reduction. All methods are equipped with the same group division and aggregation strategy. As presented in the upper half of Table 2, compared to vanilla LLM-Pruner, we have reduced the performance gap between the pruned model and the original model by 22.96% without fine-tuning, which is 29.15% when fine-tuning is applied. To ensure fairness, both models are fine-tuned on the Alpaca dataset for only 2 epochs with LoRA. Essentially, CoNNect enhances the LLM-Pruner's framework with an extra consideration of connectivity, providing good results. The results differ significantly from those produced by randomly removing parameter groups, yet the grouping strategy helps prevent the harmful effects typically associated with random pruning. However, $L_2$ regularization even results in incorrect pruning choices, which is consistent with the conclusions drawn in the previous two subsections. We show similar results for pruning 40% of the parameters through removing 50% of the parameter groups in the lower half of Table 2. Please refer to Appendix C.4 for detailed experimental settings. Moreover, we provide results for LLM-Pruner and CoNNect on LLaMA-13B; please refer to Appendix D.4 for further details.

Table 2: Zero-shot performance of the compressed LLaMA-7B. High scores are better, except for WikiText2 and PTB (indicated by the downward arrow). The bold values indicate the best results. The average is calculated among seven classification accuracies. An asterisk denotes that performance normalization is not available. The evaluation is conducted following the prompting of LLM-pruner (Ma et al., 2023). For post-training, both models are fine-tuned on the Alpaca dataset for only 2 epochs.

| Pruned Model | Method | WikiText2↓ | PTB↓ | BoolQ* | PIQA | HellaSwag | WinoGrande* | ARC-e | ARC-c | OBQA | Average |
|---|---|---|---|---|---|---|---|---|---|---|---|
| Ratio = 0% | LLaMA-7B | 12.62 | 22.15 | 73.15 | 77.48 | 73.01 | 67.09 | 52.57 | 41.47 | 42.40 | 61.02 |
| Ratio = 20% w/o tune | $L_1$ | 179.72 | 311.75 | 50.15 | 61.26 | 43.26 | 52.49 | 36.15 | 26.88 | 31.00 | 43.03 |
| | $L_2$ | 580.15 | 1022.17 | 59.66 | 57.40 | 37.07 | 52.09 | 32.53 | 28.41 | 29.80 | 42.42 |
| | Random | 22.54 | 40.10 | 46.21 | 70.46 | 59.39 | 56.51 | 41.46 | 31.91 | 37.20 | 49.02 |
| | LLM-Pruner | 19.09 | 34.23 | 57.13 | 75.08 | 66.83 | 59.75 | 50.13 | 36.35 | 39.80 | 55.01 |
| | CoNNect | **18.91** | **33.25** | **61.65** | **75.63** | **67.73** | **61.56** | **50.38** | **36.95** | **40.80** | **56.39** |
| Ratio = 20% w/ tune | $L_1$ | 24.32 | 42.85 | 59.05 | 75.24 | 65.51 | 61.56 | 47.10 | 37.37 | 39.20 | 55.00 |
| | $L_2$ | 24.75 | 42.11 | 62.72 | 75.03 | 65.17 | 63.22 | 46.17 | 36.86 | 39.80 | 55.57 |
| | Random | 19.28 | 32.92 | 54.31 | 73.18 | 64.45 | 59.91 | 47.94 | 35.15 | 40.60 | 53.65 |
| | LLM-Pruner | 17.66 | 30.51 | 65.20 | **76.88** | 68.65 | 63.93 | 52.31 | 37.03 | 40.80 | 57.83 |
| | CoNNect | **17.18** | **29.92** | **66.57** | 76.82 | **69.42** | **64.72** | **53.24** | **39.16** | **41.40** | **58.76** |
| Ratio = 40% w/o tune | $L_1$ | 888.08 | 1014.22 | 53.73 | 51.31 | 26.90 | 50.20 | 28.16 | 26.37 | 30.80 | 38.21 |
| | $L_2$ | 13783.81 | 27844.06 | 42.69 | 52.01 | 28.29 | 51.46 | 27.36 | 25.85 | 29.80 | 36.78 |
| | Random | 100.42 | 133.56 | 40.00 | 57.29 | 36.00 | 50.12 | 32.83 | 25.77 | 31.00 | 39.00 |
| | LLM-Pruner | 48.09 | 105.24 | 58.90 | 64.74 | 47.58 | **53.20** | 37.75 | 29.44 | 35.00 | 46.66 |
| | CoNNect | **46.43** | **95.08** | **60.95** | **67.30** | **50.04** | 52.09 | **38.30** | **29.86** | **36.80** | **47.91** |
| Ratio = 40% w/ tune | $L_1$ | 42.44 | 65.60 | 44.50 | **71.87** | 50.22 | 52.33 | 43.86 | 32.51 | 36.40 | 47.38 |
| | $L_2$ | 44.91 | 67.16 | 47.34 | 71.60 | 50.60 | 54.38 | 43.35 | 32.25 | 36.80 | 48.05 |
| | Random | 37.82 | 58.12 | 54.95 | 67.36 | 48.61 | 55.25 | 43.69 | 30.29 | 33.20 | 47.62 |
| | LLM-Pruner | 27.62 | 48.28 | 59.97 | 71.38 | 56.21 | **59.35** | 44.53 | 32.42 | 36.20 | 51.44 |
| | CoNNect | **27.13** | **47.44** | **61.59** | 71.06 | **57.78** | 58.48 | **45.58** | **32.85** | **39.00** | **52.33** |

# 5 Conclusions & Future Work

In this work, we introduce a novel regularizer, called CoNNect, that leverages network connectivity to promote NN sparsity. Theoretically, we prove that CoNNect is a well-behaved regularizer and aligns with the minimization of the $L_0$ norm. Through numerical experiments, we have shown that CoNNect can be effectively applied as a regularizer during training and so outperforms standard $L_1$ regularization. Moreover, we demonstrated how CoNNect can be applied competitively in a one-shot pruning framework for post-training pruning CNNs and LLMs, such as DepGraph (Fang et al., 2023), and LLM-pruner (Ma et al., 2023), showing improved results. Future work directions are ample, for example, one can extend CoNNect and make it suitable for the pruning of Recurrent Neural Networks (RNNs). Moreover, one can push CoNNect towards the semi-structured pruning paradigm, for example, as in Sun et al. (2024).

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

# Appendix

## Table of Contents

# A    Proofs

## A.1    Proof Theorem 3.1

Let $\Gamma_{i,m}$ denote the set of paths in the neural network that go from some input node $i \in V_1$ to the output node $m \in V_K$, where

$$\gamma = ((i,j),(j,k),\ldots,(l,m)) \in \Gamma_{i,m}$$

is a sequence of edges from the input layer to the output layer. Using that $\varphi^{tot}(W)$ is the sum of weights of paths from the input to the output layer (Neyshabur et al., 2015), we rewrite

$$\varphi^{tot}(W) = \sum_{i \in V_1} \sum_{m \in V_K} \sum_{\gamma \in \Gamma_{i,m}} \prod_{k=1}^{K-1} (\theta(W))_{\gamma_k} = \sum_{i \in V_1} \sum_{m \in V_K} \sum_{\gamma \in \Gamma_{i,m}} \prod_{k=1}^{K-1} \frac{|W_{\gamma_k}|}{\sum_{(r,c) \in E_k} |W_{r,c}|},$$

where $\gamma_k$ refers to the $k$th edge in a sequence $\gamma$. Then, to minimize $R(W)$, i.e., maximize $\varphi^{tot}(W)$, we need to allocate all the mass to a single path from the input to the output, which means selecting a specific sequence of weights that maximizes the product along that path, effectively minimizing the contributions from all other paths.

To show the upper bound of $|V_1| + |V_K| + K - 3$ non-zero weights in $W^*$, assume w.l.o.g. some $W^*$ where a single path $\Gamma_{i,m}$ has all mass in the network. It follows that $\varphi^{tot}(W^*) = 1$. Now, let $W'$ denote a solution where some mass from the first weight $W_{i,j}$, for $(i,j) \in \Gamma_{i,m}$ is shifted to any other weight(s) $W_{l,j}$ (note that $j$ is fixed), where $l \in V_1$ connects to $j \in V_2$. It is easily seen that $\varphi^{tot}(W') = 1$ since

$$\varphi^{tot}(W') = \sum_{l \in V_1} (\theta(W'))_{l,j} \sum_{\gamma \in \Gamma_{j,m}} \prod_{k=1}^{K-1} (\theta(W'))_{\gamma_k}$$

$$= \sum_{l \in V_1} \frac{|W'_{l,j}|}{\sum_{(r,c) \in E_1} |W'_{r,c}|} \sum_{\gamma \in \Gamma_{j,m}} \prod_{k=1}^{K-1} (\theta(W'))_{\gamma_k} = \sum_{l \in V_1} \frac{|W'_{l,j}|}{\sum_{(r,c) \in E_1} |W'_{r,c}|} \cdot 1 = 1.$$

In words, $\varphi^{tot}(W)$ is indifferent in how many of the $|V_1|$ input nodes connect to a single node in the second layer. Note that a similar argument can be made for the weights connecting the $K-1$th layer with the $K$th layer. It follows that the number of non-zero weights for $W^*$ is upper bounded by $|V_1|$ for the first layer, $|V_K|$ for layer $K-1$, and $K-3$ for the weights of the remaining layers. The resulting upper bound is then $|V_1| + |V_K| + K - 3$.

## A.2    Proof Lemma 3.2

We prove this by induction using the necessary and sufficient system of equations for stationarity in $\varphi^{tot}(W)$, see Equation (5). Assume any neural network of arbitrary size with $K = 2$ layers. Note that for this specific case any weight allocation will be stationary in $\varphi^{tot}(W)$. Now, assume a weight allocation such that $a_{\cdot i} = a_{\cdot j}$, for all $i,j \in \arg\max_{k \in V_2} a_{\cdot k}$, since adding a layer $V_{K+1}$ implies that this condition must hold to satisfy Equation (5) in the next step.

Now we add a new layer of arbitrary size $V_{K+1}$. In case $V_{K+1}$ is the last layer, it is sufficient to allocate $(\theta(W))_{i,j} > 0$, for all $i \in \arg\max_{k \in V_K} a_{\cdot k}$ to obtain a stationary point. In case the neural network is expanded with another layer $V_{K+2}$ in a next step, we let $(\theta(W))_{i,j} > 0$ for $i \in \arg\max_{k \in V_K}$ and $j \in \arg\max_{k \in V_{K+1}} a_{\cdot k}$, such that $a_{\cdot i} = a_{\cdot j}$, for all $i,j \in \arg\max_{k \in V_{K+1}} a_{\cdot k}$ to satisfy Equation (5). Note that this immediately implies $(\theta(W))_{i,j} = (\theta(W))_{r,c}$, for all $(i,j),(r,c) \in \arg\max_{(i,j) \in E_{K+1}} a_{\cdot i} a_{\cdot j}$. Hence, $(\theta(W))_{\gamma'_k} = (\theta(W))_{\gamma''_k}$, for all $k = 2,\ldots,K-2$, for all paths $\gamma$ with positive path weight.

It remains to be shown that stationarity cannot be induced by the reparameterization $\theta(W)$. To see this, we first observe that the normalization in Equation (2) is separable for each layer $k = 1,\ldots,K-1$. Simply inspecting a single layer $k$, note that $\nabla_{\theta_k} J(\theta) \neq 0$, where $\theta_k = (\theta_{i,j})_{(i,j) \in E_k}$. Moreover, let $W_k = (W_{i,j})_{(i,j) \in E_k}$ and so $\nabla_{W_k} \theta_k$, is full rank (except at $W = 0$) and thus preserves the non-zero property through the chain rule.

# B    CoNNect Implementation Details

In this section, we outline how $\varphi^{tot}(W)$ can be efficiently computed using a slightly modified forward pass of the neural network and a vector of ones as input. Below, we outline how different modules are treated in this modified forward pass. The ability to handle these modules enables the application of CoNNect across a broad spectrum of neural network architectures.

**Linear Layers:** This includes both dense (fully connected) layers and convolutional layers. The weights of these layers define the primary connections between nodes and we normalize their weights via Equation (2). The biases, however, merely shift activations (which we will exclude), and do not influence the connectivity structure and are therefore excluded.

**BN Layers:** Batch normalization layers apply standardization and scaling to the outputs of preceding layers. For the purposes of connectivity analysis, the standardization can be disregarded as it does not alter the structure of connections, but rather rescales values. Thus, we consider BN layers as identity mappings with preserved connectivity.

**Activation Functions:** Non-linear activation functions such as ReLU, sigmoid, or tanh are ignored. These functions transform node outputs but do not influence the underlying connectivity. Ignoring them simplifies the analysis without affecting the structural representation.

**Pooling Layers:** Max-pooling layers are replaced with average pooling layers. This change ensures that all input connections are treated equally in the computation of connectivity, rather than prioritizing the strongest signal as in max-pooling.

**Dropout:** Dropout layers are designed to randomly disable connections during training as a regularization method. Since they are stochastic and transient, they are ignored for connectivity analysis, as they do not represent fixed structural linkages in the network.

**Identity Connections:** Identity connections, such as skip connections in residual networks, are (generally) not parameterized and therefore can be ignored when optimizing the neural network's connectivity. Thus, we omit the identity connection in the forward pass.

## C   Experimental Settings

**Platform**: All experiments were performed on a single NVIDIA RTX4090 GPU with 24GB of memory.

### C.1   Experimental Settings for Section 4.1.1

All models have been trained to solve Equation (7), with coefficients as in Table 3. $\mathcal{L}(\hat{y}, y)$ is the Binary Cross Entropy between target and input probabilities, and $\|W\|_{2,1}$ is the often-applied $L_2$ regularization (weight decay). All models were trained for 200 epochs using Adam with a learning rate of 0.01, a cosine annealing scheduler, and batch size 256. After training, we pruned 96% of the weights in each layer using the pruning strategies discussed in Section 3.3.1: i) magnitude pruning, and ii) SynFlow pruning. Finally, the model is fine-tuned with the same hyperparameters but with a decreased initial learning rate of 0.001 for 50 epochs.

Table 3: Regularizer coefficients.

| REGULARIZER | $\lambda_1$ | $\lambda_2$ | $\lambda_3$ |
|---|---|---|---|
| NONE | 0 | 0 | $5 \times 10^{-4}$ |
| $L_1$ | $1 \times 10^{-3}$ | 0 | $5 \times 10^{-4}$ |
| CONNECT | 0 | $1 \times 10^{-1}$ | $5 \times 10^{-4}$ |

**Remark:** Synflow is traditionally introduced as a pre-training pruning method, its data-agnostic nature makes it less effective in this context, given the presence of uninformative input nodes. Moreover, SynFlow is generally regarded as a global pruning strategy. However, we frequently observed layer collapse under this configuration. In contrast, applying a local pruning approach yielded significantly better results, particularly for models without regularization and $L_1$ regularization. We thus show the results using a local pruning approach.

### C.2   Experimental Settings for Section 4.1.2

All models were trained for 300 epochs using Adam with a learning rate of 0.005. We used a linear warmup of 10 epochs with start factor 0.01 and end factor 1. Subsequently, we used a cosine annealing scheduler for the remaining 290 epochs. Fine-tuning is performed similarly, but with a learning rate of 0.0005. Regularization coefficients used are shown in Table 4, and boldfaced parameters are best performing, thus shown in Figure 4. The results for the remaining regularization coefficients are shown in figures 8 and 9, see Appendix D.2.

Table 4: Regularizer coefficients used in GNN pruning. Boldfaced parameters are used in Figure 4.

| REGULARIZER | $\lambda_1$ | $\lambda_2$ | $\lambda_3$ |
|---|---|---|---|
| NONE | 0 | 0 | $\{\mathbf{10^{-3}}, 10^{-4}\}$ |
| $L_1$ | $\{10^{-2}, \mathbf{10^{-3}}, 10^{-4}, 10^{-5}\}$ | 0 | $\{10^{-3}, \mathbf{10^{-4}}\}$ |
| CONNECT | 0 | $\{10^1, \mathbf{10^0}, 10^{-1}, 10^{-2}\}$ | $\{\mathbf{10^{-3}}, 10^{-4}\}$ |

### C.3   Experimental Settings for Section 4.2.1

When using CoNNect for pre-trained model pruning, we approximate connectivity by keeping all modules as they are to measure signal flow. Moreover, in our evaluation of connectivity, we use multiple inputs uniformly sampled between 0 and 1 instead of a single all-one for increased robustness.

**Dataset:** We train ResNet-56 with CIFAR-10 (Krizhevsky, 2009), a dataset with 60,000 32x32 images with 10 different classes. Each class has 6,000 images. Moreover, we used CIFAR-100 (Krizhevsky, 2009), a more challenging dataset consisting of 100 classes with 600 images per class, to train VGG-19.

### C.4 Experimental Settings for Section 4.2.2

In the current experiment, we use 10 randomly selected samples from Bookcorpus (Zhu et al., 2015) to be the calibration samples for establishing the dependency between parameters in the model and calculate the gradient for LLaMA-7B. To that end, we truncate each sample to a sequence length of 128. We set the coefficient $\lambda$ of CoNNect as $1 \times 10^5$. During fine-tuning, we utilize Alpaca (Taori et al., 2023), which comprises approximately 50,000 samples, to recover the capacity of the pruned model, which requires just 2 hours on our platform (NVIDIA RTX4090 GPU).

To determine which groups to prune, we compute importance scores for each weight in the model. CoNNect shares the same logic for computing the importance score from loss values as LLM-Pruner, but includes an additional connectivity term. The inputs used to evaluate connectivity are uniformly sampled between 0 and the vocabulary size. Then, specifically for $L_p$ pruning, we compute the importance of each group by computing the $L_p$ norm and prune the groups with the lowest importance scores. For random pruning, there is no need to compute importance scores for each group–we simply randomly select certain groups for pruning. Moreover, we leave the first three layers and the final layer unchanged (similar to Ma et al. (2023)), as substantial changes to the parameters of these layers greatly influence the performance of the model. Finally, the discovered groups within each module are pruned according to a predetermined ratio. The pruning rate for the selected groups is higher than the pruning ratio for the parameters since some layers (e.g., the excluded layers) retain their parameters. For a total of 40% parameter removal, we must prune 50% of the groups specifically from the fourth to the thirtieth layer.

**Datasets:** To assess the model performance, we conduct a zero-shot perplexity analysis on WikiText2 (Merity et al., 2022) and PTB (Marcus et al., 1993), and then follow Gao et al. (2021) to test the model with zero-shot classification tasks on common sense reasoning datasets: BoolQ (Clark et al., 2019), PIQA (Bisk et al., 2020), HellaSwag (Zellers et al., 2019), WinoGrande (Sakaguchi et al., 2021), ARC-easy, ARC-challenge (Clark et al., 2018), OpenbookQA (Mihaylov et al., 2018), where the model ranks the choices in these multiple-choice tasks.

# D   Ablation Studies

## D.1   Impact of Initializations and Regularizer Strength in Section 3.3.1

To show the robustness of our results, we conduct an ablation study to: 1) analyze the impact of different initializations, and 2) the regularization strengths.

First, we present the results for 100 different initializations, where we show the (aggregated) train and test loss in Figure 5 and the fine-tuned accuracies in Figure 6. Roughly speaking, the final accuracy for each model can be categorized by the ability to find the network connecting the input nodes 1 and 2 to the output layer. If the fine-tuned accuracy is around 0.50, the algorithm was unable to connect node 1 and node 2 to the output (see figures 3(a) and (b)). If the fine-tuned accuracy is around 0.75, the algorithm was able to connect node 1 or node 2 to the output. Finally, if the algorithm preserved the edges connecting node 1 and node 2, it found the correct network and achieved an accuracy of more than 0.95 (see Figure 3(c)).

As shown in Figure 5, CoNNect regularization via $\varphi^{tot}(W)$ is beneficial to both pruning strategies. It is noteworthy that SynFlow pruning does not offer much further improvement over connectivity regularization compared to simple magnitude pruning, although it seems to slightly decrease the frequency of layer collapse compared with magnitude pruning, see Table 5. This can be attributed to the fact that CoNNect regularization has already trained the network to use the correct paths to model the current problem, as shown in Figure 3(c). It thus suffices to apply a simple magnitude pruning to identify these paths.

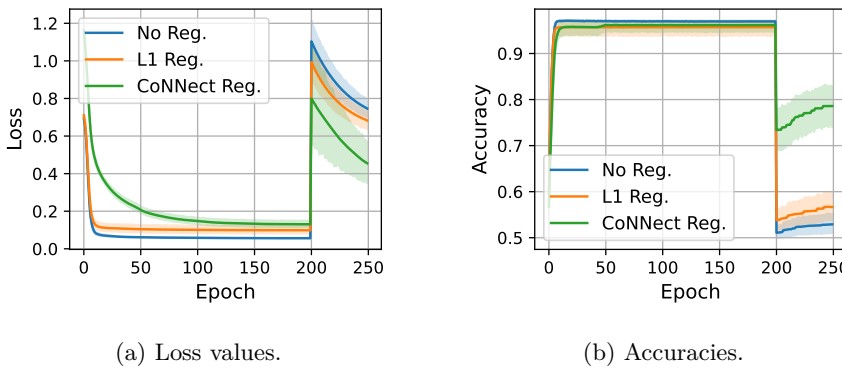

(a) Loss values.                                    (b) Accuracies.

Figure 5: Learning curves for solving Equation (4). SynFlow pruning happens at iteration 200. Bandwidths are 95% confidence intervals.

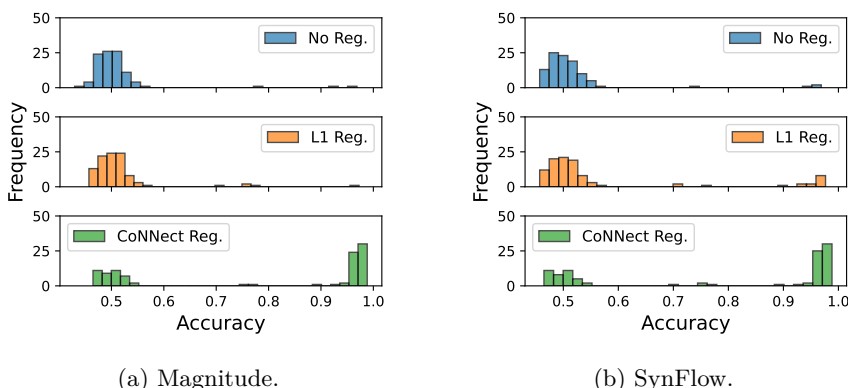

(a) Magnitude.                                    (b) SynFlow.

Figure 6: Fine-tuned accuracy after magnitude pruning and SynFlow pruning of regularized models.

Now we perform experiments with different values of $\lambda$, see Table 6 for an overview. Specifically, increasing $\lambda_1$ by one order of magnitude to 0.01 causes a frequent occurrence of layer collapse, although it does increase

Table 5: Layer collapse frequencies (out of 100 runs) per pruning method and regularization type.

| Pruning Method | No Reg. | $L_1$ Reg. | CoNNect Reg. |
|---|---|---|---|
| Magnitude | 81 | 83 | 4 |
| SynFlow | 71 | 61 | 1 |

Table 6: Regularizer coefficients used for producing figures 7(a)-(d), respectively.

| | EXPERIMENT 1 | | | EXPERIMENT 2 | | | EXPERIMENT 3 | | | EXPERIMENT 4 | | |
|---|---|---|---|---|---|---|---|---|---|---|---|---|
| REGULARIZER | $\lambda_1$ | $\lambda_2$ | $\lambda_3$ | $\lambda_1$ | $\lambda_2$ | $\lambda_3$ | $\lambda_1$ | $\lambda_2$ | $\lambda_3$ | $\lambda_1$ | $\lambda_2$ | $\lambda_3$ |
| NONE | 0 | 0 | $5 \times 10^{-4}$ | 0 | 0 | $5 \times 10^{-3}$ | 0 | 0 | $5 \times 10^{-2}$ | 0 | 0 | $5 \times 10^{-3}$ |
| $L_1$ | $1 \times 10^{-2}$ | 0 | $5 \times 10^{-4}$ | $1 \times 10^{-3}$ | 0 | $5 \times 10^{-3}$ | $1 \times 10^{-3}$ | 0 | $5 \times 10^{-2}$ | $5 \times 10^{-4}$ | 0 | $5 \times 10^{-3}$ |
| CONNECT | 0 | 1 | $5 \times 10^{-4}$ | 0 | $1 \times 10^{-1}$ | $5 \times 10^{-3}$ | 0 | $1 \times 10^{-1}$ | $5 \times 10^{-2}$ | 0 | $5 \times 10^{-2}$ | $5 \times 10^{-3}$ |

the performances for the cases without layer collapse, see Figure 7(a). Changing $\lambda_2$ by one order of magnitude to 1 did not cause any specific change, arguing for the stability of CoNNect. Moreover, increasing $\lambda_3$ by one order of magnitude to 0.005 seems to improve the model performance overall, especially for the CoNNect regularized model, see Figure 7(b). Increasing $\lambda_3$ by another order of magnitude still shows very competitive results for CoNNect. Finally, we decrease $\lambda_1$ and $\lambda_2$ to 0.0005 and 0.05 respectively, and see that the regularizers become too weak leading the results to converge toward those of standard $L_2$ regularization.

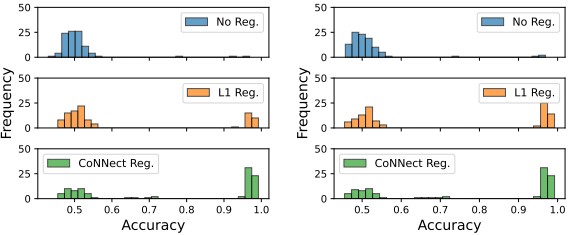

(a) Fine-tuned accuracy after magnitude and SynFlow pruning (Experiment 1).

(b) Fine-tuned accuracy after magnitude and SynFlow pruning (Experiment 2).

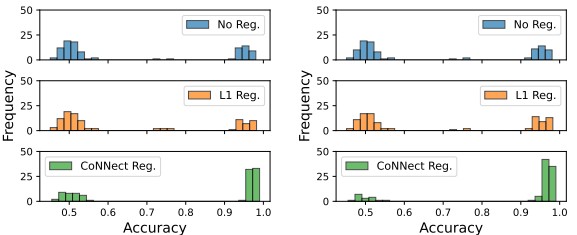
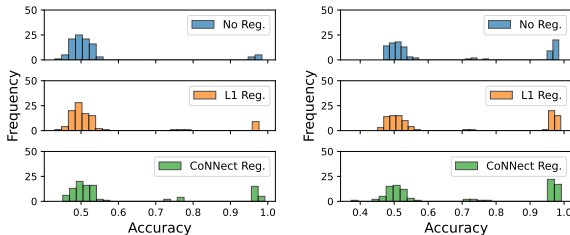

(c) Fine-tuned accuracy after magnitude and SynFlow pruning (Experiment 3).

(d) Fine-tuned accuracy after magnitude and SynFlow pruning (Experiment 4).

Figure 7: Fine-tuned accuracy after magnitude and SynFlow pruning for different regularization settings.

## D.2 Impact of Initializations and Regularizer Strength in Section 4.1.2

Ablation on the regularization coefficients for both $L_1$ and CoNNect regularization are shown in figures 8 and 9. Note that the cases with $\lambda_1 = 0$ and $\lambda_2 = 0$ correspond to settings where neither $L_1$ nor CoNNect regularization is applied.

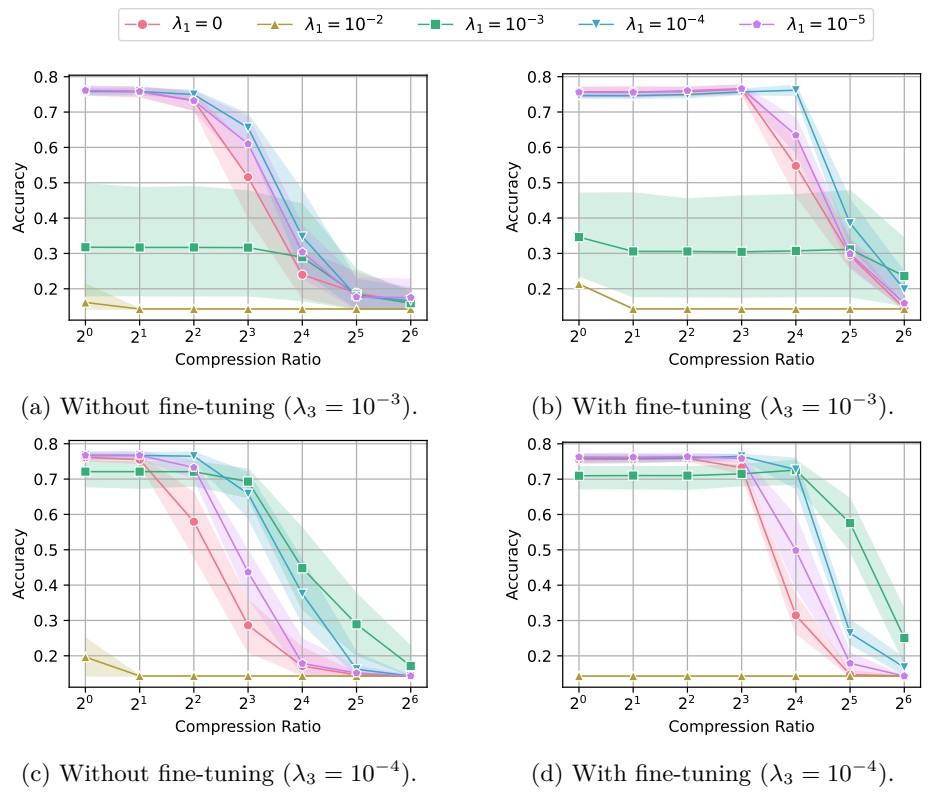

Figure 8: Accuracies of GNNs for given compression ratios under $L_1$ regularization.

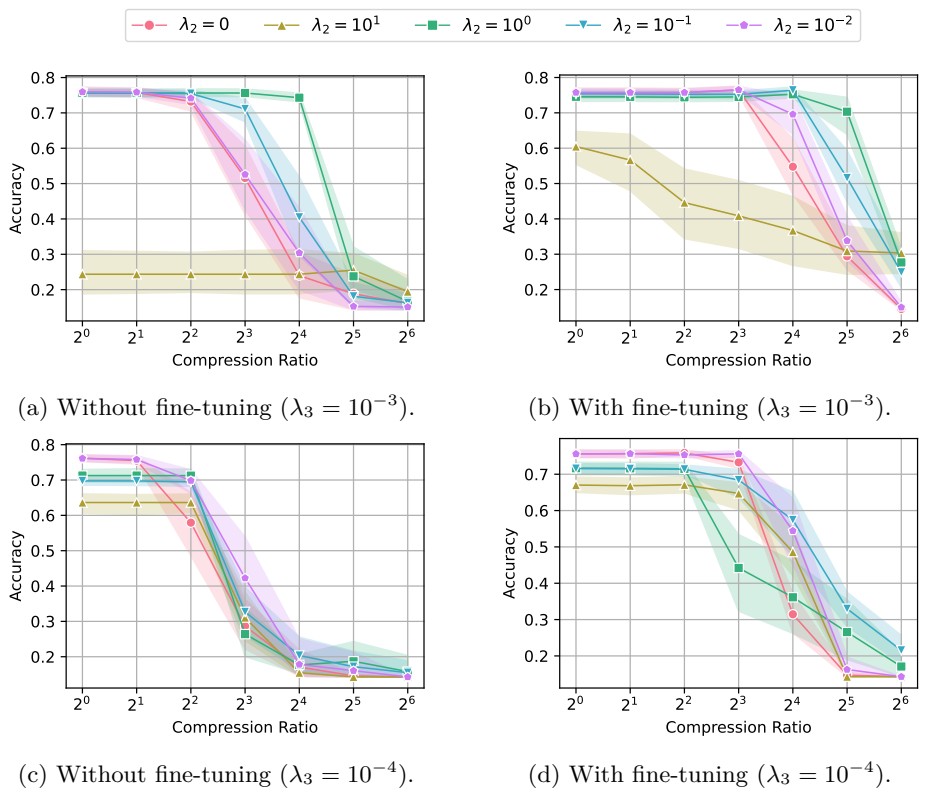

Figure 9: Accuracies of GNNs for given compression ratios under CoNNect regularization.

### D.3 Deep Weight Factor Regularization in the Numerical Example in Section 4.1.2

In this section, we present the results of applying Deep Weight Factorization (DWF) to the numerical example from Section 4.1.2 (Kolb et al., 2025). DWF introduces an $L_{D/2}$ regularization by reparameterizing the network weights $W$ using $D \geq 2$ factors. More formally, each parameterized weight is factorized as

$$W_{i,j} = \omega_{i,j}^{(1)} \times \cdots \times \omega_{i,j}^{(D)}.$$

This formulation allows $L_{D/2}$ regularization in the weight space $W$ to be conveniently implemented through weight decay on the individual factors.

Figure 10 reports the results, where we vary both the number of factors $D$ and the strength of the weight decay $\lambda_3$. It can be seen that these results are not competitive with CoNNect, shown in Figure 9.

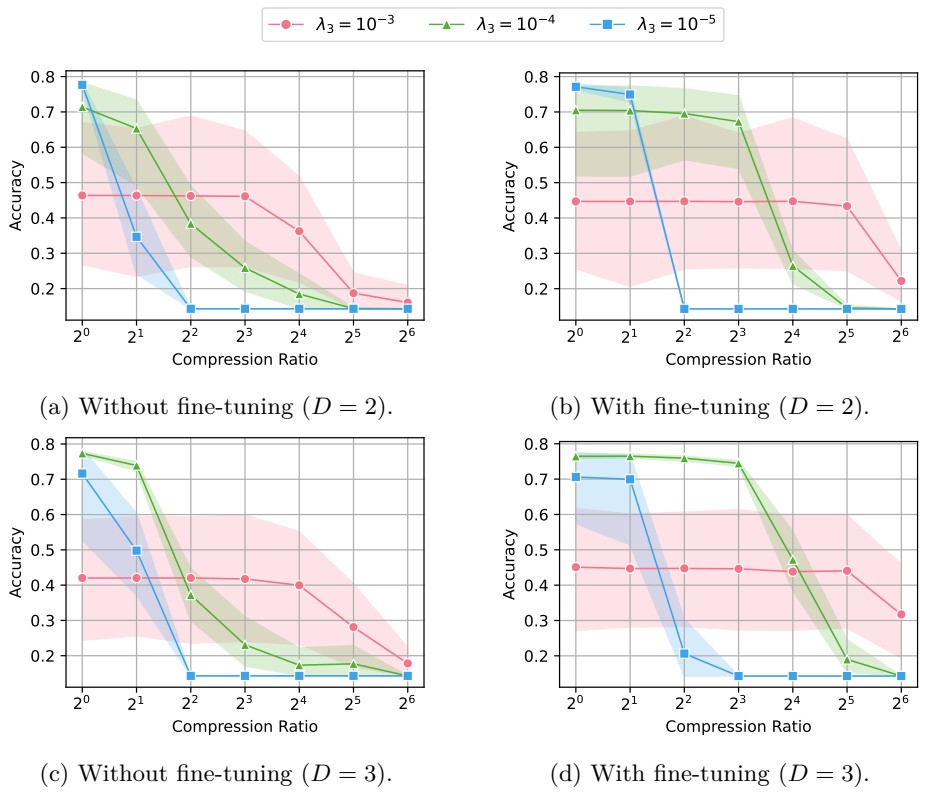

(a) Without fine-tuning $(D = 2)$.  (b) With fine-tuning $(D = 2)$.

(c) Without fine-tuning $(D = 3)$.  (d) With fine-tuning $(D = 3)$.

Figure 10: Accuracies of GNNs for given compression ratios under DWF regularization.

### D.4 Supplemental Results for LLaMA-13B

We further evaluate our performance on LLaMA-13B (Touvron et al., 2023), similar to Section 4.2.2, where 20% and 40% of the parameters are pruned. We show these results in tables 7 and 8. We note that the LLaMA-13B model used in our evaluation is community-released `yahma/llama-13b-hf`, as the original checkpoint used in Ma et al. (2023) is no longer available. This may slightly affect the baseline performance. All models are evaluated using the same test benchmarks as presented in Table 2. The results demonstrate that CoNNect almost always outperforms the baseline method across multiple benchmarks, regardless of whether fine-tuning is applied. This observation is consistent with the findings in the main text, indicating the effectiveness and robustness of our methods on larger-scale models.

Table 7: Zero-shot performance of the compressed LLaMA-13B with a 20% pruning ratio.

| Dataset | LLaMA-13B | 20% w/o tune | | 20% w/ tune | |
|---|---|---|---|---|---|
| | | LLM-Pruner | CoNNect | LLM-Pruner | CoNNect |
| WikiText2↓ | 11.58 | 16.62 | **16.30** | 15.64 | **15.16** |
| PTB↓ | 44.56 | 60.91 | **59.03** | 59.96 | **58.80** |
| BoolQ* | 68.53 | 64.43 | **69.05** | 65.69 | **72.63** |
| PIQA | 79.05 | **77.20** | 76.71 | 78.18 | **79.11** |
| HellaSwag | 76.21 | 73.45 | **73.93** | 74.99 | **75.37** |
| WinoGrande* | 70.09 | 67.56 | **68.19** | 68.82 | 67.88 |
| ARC-e | 59.81 | 56.90 | **58.21** | 57.70 | **60.98** |
| ARC-c | 44.62 | 40.10 | **40.96** | 42.06 | **43.69** |
| OBQA | 42.20 | 41.40 | **41.60** | 43.60 | 43.40 |
| Average | 62.93 | 60.15 | **61.24** | 61.58 | **63.29** |

Table 8: Zero-shot performance of the compressed LLaMA-13B with a 40% pruning ratio.

| Dataset | LLaMA-13B | 40% w/o tune | | 40% w/ tune | |
|---|---|---|---|---|---|
| | | LLM-Pruner | CoNNect | LLM-Pruner | CoNNect |
| WikiText2↓ | 11.58 | 35.18 | **32.41** | 22.49 | **21.85** |
| PTB↓ | 44.56 | 120.19 | **106.90** | 78.21 | **76.10** |
| BoolQ* | 68.53 | 62.05 | **62.11** | 62.17 | **62.48** |
| PIQA | 79.05 | **73.34** | 72.20 | 75.90 | **75.95** |
| HellaSwag | 76.21 | 59.63 | **61.82** | 66.05 | **66.64** |
| WinoGrande* | 70.09 | 55.49 | **55.96** | 61.56 | **62.12** |
| ARC-e | 59.81 | 44.19 | **45.16** | 52.31 | 52.31 |
| ARC-c | 44.62 | 33.36 | **34.04** | **36.35** | 35.92 |
| OBQA | 42.20 | 38.60 | **39.20** | 39.80 | **41.40** |
| Average | 62.93 | 52.38 | **52.93** | 56.31 | **56.69** |

# E   Computational Improvement for LLMs

In Table 9, we show the computational improvement of CoNNect integrated in LLM-pruner across different pruning ratios in terms of complexity. Indeed, speed-up is close to the compression ratio.

Table 9: Computational complexity of CoNNect-pruned LLaMA-7B model for different pruning ratios. Speed-up is the GMACs of the base model divided by the GMACs of the pruned model.

| Pruning Ratio | Parameter Count | GPU Memory (MiB) | Comp. Complexity (GMACs) | Speed-up (GMACs) |
|---|---|---|---|---|
| 0% | 6.7B | 12892.6 | 425.1 | - |
| 20% | 5.4B | 10383.7 | 340.5 | 1.3× |
| 40% | 4.1B | 7952.6 | 255.8 | 1.7× |

