# OpenReview forum: "CoNNect: Connectivity-Based Regularization for Structural Pruning of Neural Networks"
_TMLR — Accepted by TMLR_

### Review · Reviewer_425e · 2025-07-08

**Summary Of Contributions:**

This paper introduces a novel regularizer called CoNNect which results in a sparse network while ensuring connectivity between input and output layers. Furthermore, the authors demonstrate via experiments that this regularizer helps improve the results of pruning strategies.

The authors state the following two axioms for a good pruning algorithm:

1. Pruning should reduce the number of weights while minimizing the impact on accuracy and increasing computational efficiency
2. Pruning should preserve network connectivity and the flow of information from input to output

The paper then highlights that while many algorithms satisfy the first axiom, the second axiom is often neglected leading to problems like layer collapse with the exception of SynFlow pruning technique which is designed to preserve network flow.

The authors then introduce their CoNNect regularizer which is inspired by Katz centrality and SynFlow  which satisfies both axioms and results in a sparse network where flow of information is preserved. The paper also provides theoretical guarantees to show the robustness of their regularizer.

**Audience:**

Yes

**Claims And Evidence:**

Yes

**Requested Changes:**

I believe that the paper is strong enough to be accepted at TMLR as is, but I would still encourage the authors to address some of the comments mentioned in the weakness section.

**Strengths And Weaknesses:**

**Strengths:**

- The paper is well-written and easy to follow
- The paper addresses an important problem in pruning about network connectivity
- The paper introduces a novel regularizer and provides theoretical guarantees for its robustness
- A good set of experiments are conducted to show the efficacy of the regularizer

**Weaknesses:**

- I would like to a discussion / set of experiments on the limitations of CoNNect Regularizer. Are there any network architectures / scenarios for which this regularizer would fail?
- A comparison against Iterative Magnitude Pruning in the context of the Lottery Ticket Hypothesis would be interesting.
- A bit more discussion on future directions would be helpful.

---

> ### Author Response · Authors · 2025-07-23
> **Response on Weaknesses and Requested Changes**
>
> ### Weaknesses
>
> **(1)** Thank you for this raising point. The current definition of connectivity, as used in CoNNect, is not directly suited for recurrent neural networks due to their cyclic and temporal structure. However, we believe the concept of CoNNect can be extended to such architectures, where one has to carefully consider if and how cyclical connections contribute towards connectivity. We see this as a valuable direction for future research to make CoNNect even more broadly applicable. We have now included a discussion of this limitation in the revised manuscript.
>
>
> **(2)** Indeed, we did not include comparisons with Iterative Magnitude Pruning (IMP), which is known to yield highly accurate sparse networks. However, IMP relies on multiple rounds of retraining, making it substantially more computationally intensive (e.g., see [1]). In contrast, all methods evaluated in our paper - including CoNNect - do not require repeated retraining cycles. For this reason, we chose to focus our comparisons on methods with similar computational cost, to ensure a fair and practical evaluation setting. But we note that CoNNect is in principle compatible with multi-step schemes like IMP, and may potentially benefit from more computational consumption.
>
>
> **(3)** Thank you for the suggestion. We have expanded the section on future work. This now covers extending the definition of connectivity to accommodate recurrent neural networks, as well as exploring the application of CoNNect within the semi-structured pruning paradigm, which is not addressed in the current manuscript. For the latter, we already have promising results, please see our response at Weakness 2 for reviewer 29EU, but as to keep the paper within the scope of structural pruning, we have not incorporated these in the revision. We would sincerely appreciate the reviewer's input on whether the inclusion of this semi-structured pruning evaluation would enhance the final version of the paper.
>
>
> ### Requested Changes
>
> **(1)** Thank you for the valuable feedback. We have incorporated the suggested revisions into the updated manuscript accordingly.
>
> > References
> > [1] Choi, Moonseok, Hyungi Lee, Giung Nam, and Juho Lee. "Sparse weight averaging with multiple particles for iterative magnitude pruning." arXiv preprint arXiv:2305.14852 (2023).

---

> ### Author Response · Authors · 2025-08-08
>
> Dear reviewer 425e,
>
> We would like to thank you once again for your positive feedback, as well as for the time and effort you have devoted to helping improve the manuscript. If there is any remaining feedback or points requiring clarification, please do not hesitate to let us know. We sincerely look forward to your response.
>
> Best,
> Authors

---

### Review · Reviewer_29EU · 2025-07-15

**Summary Of Contributions:**

This paper proposes a regularization of the pruning methodology based on connectivity. The authors propose a method of limiting the removal or reducing to 0-weight such connections in a network that would result in network that is not connected, i.e. the input cannot be transmitted to output. The paper is well written, and the evaluation is performed on several different models and datasets. The proofs of the theorems seem to be also correct.

In more details, the authors use the standard optimization notation where the proposed method is introduced as a penalty for dropping certain weights. Then by comparing to other regularizations the authors show that for certain cases their proposed regularization improves lightly the result when other regularizes are used.

**Audience:**

Yes

**Claims And Evidence:**

Yes

**Requested Changes:**

More experiments on unstructured pruning. While the authors provided some experiments, for the present paper the experimental data is not enough. This is mainly due to the fact that the paper is of very applied nature and empirical data of the regularization should be well understood before propagated. While the theoretical results seem promising the authors should make a special case to in details evaluate major models of neural networks in order to observe its behavior.

**Strengths And Weaknesses:**

The main strength of this paper is relative simplicity and the claims of evidence. This paper show that the proposed regularization works for several different models and in general it is very intuitive and efficient approach.

The weaknesses are several:

First the authors claim that the other regularization methods such as $l_1$ or $l_2$ can result in network or layer collapse. However, during the evaluation of the proposed regularization not a single time such case was demonstrated. This can be seen on relatively similar results during the numerical evaluation. If the layer collapse would be occurring, wouldn't it be more visible or visible in at least one case?

The second issue is with the evaluation of the unstructured pruning. The authors show only very small toy results and do not investigate any further. Again, it would be very desirable to have an illustration of the advantages of the proposed regularizer with examples of layer collapse and real-world experimental data.

Moreover, the authors did not mention multi step pruning or retraining after pruning. Is there a reason for this?

---

> ### Author Response · Authors · 2025-07-23
> **Response on Weaknesses and Requested Changes (1/2)**
>
> ### Weaknesses
>
> **(1)** We thank the reviewer for raising this concern. To address it, we have made the effects of layer collapse more explicit in the revised manuscript. Figure 3 already presents visual demonstrations of layer collapse under both $L_2$ and $L_1$ regularization. Additionally, we have added results on the frequency of layer collapse for the MLP toy example in Appendix D.1, where $L_2$ and $L_1$ regularization show high frequencies of layer collapse, whereas for CoNNect this is not the case. Moreover, layer collapse also occurs at extreme compression ratios ($2^6$) for No Regularization and $L_1$ Regularization in the GNN experiment in Figure 4, whereas this does not occur for CoNNect Regularization.
>
> This demonstrates that layer collapse indeed occurs more frequently when using $L_2$ and $L_1$ regularization compared to our proposed CoNNect regularizer. Furthermore, we emphasize that layer collapse is not limited to unstructured pruning. We show that it can also occur in structured pruning - specifically channel pruning - when applying global pruning strategies that may remove all channels in a layer.
>
> Importantly, our results also indicate that the improved performance observed in Figure 4 cannot be solely attributed to avoiding layer collapse, as no collapse occurs at lower compression ratios. In this regime, the CoNNect regularizer still identifies more promising channels than $L_1$ and $L_2$ regularization. This highlights that our method not only prevents collapse but also facilitates more effective sparse subnetwork discovery overall.
>
> **(2)** We appreciate the reviewer’s interest in a more extensive evaluation of unstructured pruning. Our paper focuses primarily on structured pruning, which we believe represents the most concept-relevant and strict setting. Structured pruning enables true computational acceleration without relying on hardware or software-specific sparsity support, by systematically removing larger functional modules (e.g., channels or blocks). However, the CoNNect framework treats the network as a collection of modular components and can be applied at arbitrary granularity - including individual parameters - thus naturally covering unstructured pruning as a special case. The small unstructured toy example in the main body is included to illustrate, in a visually intuitive manner, the potential failure modes that arise when connectivity is not enforced.
>
> To further demonstrate CoNNect’s potential in accelerating model inference, we integrate CoNNect into a different pruning paradigm, semi-structured pruning, which enables actual speedup under specific hardware support. As shown below, we use the recently popular and widely adopted Wanda [1] framework and conducted experiments on LLaMA-7B and LLaMA-2-7B with 2:4 and 4:8 sparsity patterns. CoNNect consistently demonstrates an improvement in perplexity across different calibration set sizes, particularly in low-data regimes:
>
> **Mean (STD) perplexity on Llama-7b.**
> |Pruning|Method|1|4|16|64|128|
> |:-|:-|:-|:-|:-|:-|:-|
> |2:4|Wanda|12.46 (0.23)|11.98 (0.20)|11.71 (0.33)|11.62 (0.08)|11.56 (0.06)|
> | |CoNNect|**12.22 (0.24)**|**11.68 (0.14)**|**11.36 (0.24)**|**11.43 (0.13)**|**11.23 (0.05)**|
> |4:8|Wanda|9.05 (0.16)|8.81 (0.04)|8.74 (0.06)|**8.69 (0.00)**| **8.63 (0.04)**|
> | |CoNNect|**8.91 (0.14)**|**8.67 (0.02)**|**8.64 (0.04)**|8.71 (0.02)|**8.63 (0.03)**|
>
> **Mean (STD) perplexity on Llama-2-7b.**
> |Pruning|Method|1|2|8|16|32|
> |:-|:-|:-|:-|:-|:-|:-|
> |2:4|Wanda|12.48 (0.46)|11.99 (0.32)|11.73 (0.32)|11.55 (0.14)|11.48 (0.11)|
> | |CoNNect | **11.66 (0.33)**| **11.40 (0.30)**| **11.28 (0.25)**|**11.22 (0.11)**|**11.09 (0.06)**|
> |4:8|Wanda|8.41 (0.13)|8.29 (0.12)|8.22 (0.11)|8.17 (0.06)|8.14 (0.04)|
> | |CoNNect|**8.18 (0.12)**|**8.05 (0.10)**|**8.04 (0.09)**|**8.06 (0.05)**|**8.01 (0.03)**|
>
> These results further confirm the practical utility of connectivity regularization - even in other pruning paradigms - and suggest broader applicability of CoNNect across the pruning spectrum.
>
> Note that we have not included these results in the revision, as the current work mainly focusses on structural pruning. Nonetheless, if preferred by the reviewer, we can include these results in the main text and appreciate the guidance of the reviewer.

---

> > ### Author Response · Authors · 2025-07-23
> > **Response on Weaknesses and Requested Changes (2/2)**
> >
> > **(3)** We thank the reviewer for raising this point. Regarding multi-step pruning, although our main text focuses on one-shot pruning for clarity and efficiency, the LLM pruning pipeline presented in Section 4.2.2 in fact supports iterative pruning. Specifically, after each pruning step, the importance scores of the remaining parameter groups are re-evaluated based on updated connectivity and loss gradients, and used for the next pruning round. While we did not emphasize this in the manuscript, we confirm that CoNNect is fully compatible with multi-step iterative pruning schemes, should the use case demand it.
> >
> > As for retraining, we interpret this as fine-tuning the model after pruning. If this is the reviewer’s intended meaning, we would like to clarify that all of our pruning results include a post-pruning retraining phase, labeled as 'with fine-tuning' in the manuscript. In fact, the performance gains from CoNNect are consistently visible both with and without fine-tuning, as shown in our results. This highlights that CoNNect not only facilitates more effective pruning decisions but also produces pruned models that are inherently better conditioned - even before any retraining.
> >
> > ### Requested Changes
> >
> > **(1)** We appreciate the reviewer’s point about the importance of evaluating the empirical behavior of the proposed regularizer. While our main focus is on structured pruning, due to its practical relevance for hardware-agnostic acceleration and improving computational efficiency (Axiom 1) [2], we fully acknowledge the significance of other pruning paradigms in many research contexts. To that end, we conducted additional semi-structured pruning experiments during the rebuttal period. Specifically, we integrated CoNNect into a recent and widely adopted semi-structured pruning framework and evaluated it on LLM pruning tasks. As shown in the tables above, CoNNect consistently improves perplexity across a range of calibration sizes, particularly in low-data regimes. Since the original manuscript aims to maintain a clear and coherent focus on structured pruning, we have not yet incorporated these additional results into the revision. However, we would greatly value the reviewer’s guidance on whether including this semi-structured pruning evaluation would help strengthen the final version of the paper.
> >
> > In addition, we would like to highlight that we have already updated the main paper to include new results demonstrating layer collapse under various pruning strategies, see Appendix D.1. These visualizations and quantitative measurements (e.g., frequency of collapse) have been added to better illustrate the practical impact of connectivity-aware regularization. We kindly invite the reviewer to refer to the updated sections in the revised manuscript for further details.
> >
> > > References:
> > > [1] Sun, Mingjie, Zhuang Liu, Anna Bair, and J. Zico Kolter. "A Simple and Effective Pruning Approach for Large Language Models." In Proceedings of the International Conference on Learning Representations (ICLR) 2024, 2024.
> > > [2] Cheng, Hongrong, Miao Zhang, and Javen Qinfeng Shi. "A survey on deep neural network pruning: Taxonomy, comparison, analysis, and recommendations." IEEE Transactions on Pattern Analysis and Machine Intelligence (2024).

---

> > > ### Author Response · Authors · 2025-08-08
> > >
> > > Dear reviewer 29EU,
> > >
> > > We would like to thank you once again for your valuable comments on our submission.
> > >
> > > As the discussion phase is approaching its conclusion, we would like to kindly confirm whether we have sufficiently addressed all of your concerns (or at least a substantial part of them). With your feedback, we can proceed with a final revision of the paper, incorporating your suggestions along with the final remarks from the other reviewers.
> > >
> > > If there are any remaining questions or points requiring further clarification, please do not hesitate to let us know. We sincerely look forward to your response.
> > >
> > > Best,
> > > Authors

---

### Review · Reviewer_s6Vh · 2025-07-16

**Summary Of Contributions:**

This paper presents a regularization function that leads to sparse neural networks while keeping them connected to, ultimately, prune the networks while avoiding layer collapse (i.e., that a layer/neurons produce features that the following layer does not use). The regularizer is evaluated on an small MLP, on a graph neural network, and on pretrained CNN and LLM where it achieved higher performance than other methods.

**Audience:**

Yes

**Broader Impact Concerns:**

This method (a strategy to improve pruning in deep learning networks) has no direct negative societal impact.

**Claims And Evidence:**

No

**Requested Changes:**

# Clarifications:
- I have the impression that the issue that this regularizer aims to solve occurs only in unstructured pruning because, in structured pruning (filter or layer pruning), you can eliminate filters/layers directly without the issue of neurons/filters/layers producing features that won't be utilized. This is because, if you eliminate a filter in layer X, then, you also need to adjust layer X+1. Consider layer X being 32x1x3x3, and layer X+1 being 64x32x3x3; if you eliminate 10 filters from layer X, you will have that layer X is 22x1x3x3 and layer X+1 must be adjusted to 64x22x3x3. If my impression is correct (i.e., that this method is specifically for unstructured pruning in order to "achieve" structured pruning) then, I didn't find it too clear, and I suggest clarifying it.
- In connection to my previous comment: doesn't "Axiom 2" hold in filter pruning? I can't see how, in filter pruning in CNNs, you would end up having this layer collapse issue.
- After reading the Related work, it is unclear how redundancy elimination methods, such as [1,2,3] fit into this classification of previous work.
- Could you, intuitively, indicate why minimizing the regularizer would lead the models to have a "structured sparsity" allowing structured pruning?

[1]: Sauron U-Net: Simple automated redundancy elimination in medical image segmentation via filter pruning. Pattern Recognition 2024

[2]: Centripetal sgd for pruning very deep convolutional networks with complicated structure. CVPR 2019

[3]: Filter pruning via geometric median for deep convolutional neural networks acceleration. CVPR 2019

# Small modifications/comments
- "Layer collapse" is introduced in the abstract, but it's not defined. For the reader who doesn't know it, it would be beneficial to either omit this term and talk about what is it.
- 1. Introduction "Pruning weights based on absolute values". What does this exactly refer to? Based on the L norms?
- The equation above Eq. 2: Please indicate what's "k". I guess this is the layer.
- Eq. 2: Isn't $W_{i,j}$ a single value (the edge connecting neuron $i$ with $j$)? If so, why doing $|.|$?
- Below Eq. 3 "[...] exclusively influences the learnable structure". I think that rather than "structure", it is meant "parameter".
- Below Eq. 3 "sum of all (multiplicative) reparameterized weighted". I think that rather than "reparameterized, it is meant "normalized"?
- Below Eq. 3. Typo "paramterized".
- Below Eq. 3. "Moreover, ..." does this statement mean that one needs to replace all biases all the time?
- Page 6. "Also, our regularizer does not force.." I found this sentence a bit convoluted.

**Strengths And Weaknesses:**

# Strengths
- The manuscript is easy to follow.
- The proposed regularizer is clearly defined.
- This method permits to utilize unstructured pruning approaches without the issue of layer collapse, enabling their efficient use.
- Experiments were conducted on a number of neural network architectures.


# Weaknesses
- Unintuitive regularizer. It is unclear why minimizing the layer-wise normalized weights (in essence, Eq. 2) would lead to a sparsity that respects the network's structure and would allow to do structure pruning.
- Very few comparisons with current work.
- Confusing/fuzzy categorization of Related work. Related work is divided into "Regularization-based pruning" and "Stage-based pruning": Regularization-based are methods that minimize a regularizer, and stage-based are those that explicitly remove the parameters. However, what if a method minimizes a regularizer and also explicitly removes the parameters?
- No code is shared. I consider this an important weakness that, if addressed, could improve the impact of this work.

---

> ### Author Response · Authors · 2025-07-23
> **Response on Weaknesses and Requested Changes (1/2)**
>
> ### Weaknesses
>
> **(1)** Thank you for your comment. We agree that the intuition behind the regularizer may not be immediately clear. In Section 3.3.1, we introduce the regularizer at the weight level, as this is the most granular and intuitive setting to explain the underlying mechanism. Theorem 3.1 demonstrates how the regularizer operates in this setting by encouraging sparsity through maximization.
>
> We then extend this principle in Section 3.3.2 to structural pruning by applying the regularizer to scaling factors associated with higher-level structures such as channels. These scaling factors act as inserted linear layers composed of individual scalar weights per channel, so the insights from the weight-level analysis (Theorem 3.1) naturally transfer. As a result, CoNNect encourages the network to rely only on the most important channels.
>
> We deliberately start with the weight-level formulation to help the reader understand that the same principle generalizes to any level of granularity - including blocks or channels. In fact, individual weights can also be viewed as candidate modules for pruning, where each weight can be interpreted as the product of a scaling factor and an underlying base weight, making the transition to coarser structures seamless. Presenting the regularizer directly at the channel level would obscure this flexibility, so we chose a bottom-up exposition for clarity.
>
> **(2)** To further demonstrate CoNNect’s capabilities, we incorporated the Deep Weight Factorization (DWF) parameterization [1] in our GNN experiment (Section 4.1.2) for comparison with recent methods. DWF applies an $L_{2/D}$ regularization by reparameterizing the network weights using $D$ factors.
>
> As shown in the table below and in Figure 4, DWF does not outperform CoNNect. We are happy to include these results in the main paper upon the reviewer's request. Additionally, our implementation of DWF has been included in the updated supplementary material.
>
> **Mean with 98% CI accuracy for Deep Weight Factorization for 2 and 3 factors across compression ratios as in Figure 4 of the manuscript.**
> |Factors ($D$)|$\lambda_3$|2⁰|2¹|2²|2³|2⁴|2⁵|2⁶|
> |-|-|-|-|-|-|-|-|-|
> |2|10⁻³|0.45 (±0.32)|0.45 (±0.32)|0.45 (±0.32)|0.45 (±0.32)|0.45 (±0.32) | 0.43 (±0.31) |0.22 (±0.10)|
> |2|10⁻⁴|0.70 (±0.20)|0.70 (±0.20)|0.70 (±0.19)|0.67 (±0.19)|0.26 (±0.07)|0.15 (±0.01)|0.14 (±0.00)|
> |2| 10⁻⁵|0.77 (±0.01)|0.75 (±0.03)|0.14 (±0.00)|0.14 (±0.00)|0.14 (±0.00) |0.14 (±0.00)|0.14 (±0.00)|
> |3| 10⁻³|0.45 (±0.18)|0.45 (±0.18)|0.45 (±0.18)|0.45 (±0.18)|0.44 (±0.18)|0.44 (±0.18)|0.32 (±0.14)|
> |3| 10⁻⁴|0.76 (±0.01)|0.76 (±0.01)|0.76 (±0.01)|0.74 (±0.01)|0.47 (±0.09)|0.19 (±0.05)|0.14 (±0.00)|
> |3| 10⁻⁵|0.71 (±0.15)|0.70 (±0.14)|0.21 (±0.10)|0.14 (±0.00)|0.14 (±0.00)|0.14 (±0.00)|0.14 (±0.00)|
>
>
> **(3)** Our categorization in the Related Work section is intended to highlight two predominant paradigms in pruning. We distinguish between (1) *regularization-based pruning*, where certain weights are guided toward zero (or redundancy) through optimization via an adjustment of the descent direction. This process is followed by a straightforward removal of ineffective or redundant weights. Moreover, (2) *stage-based pruning*, which involves explicitly removing parameters at specific stages, often based on magnitude or other criteria, without necessarily altering the descent trajectory during training.
>
> We fully acknowledge that many modern pruning methods combine both paradigms - for example, incorporating regularization during training followed by hard thresholding at intervals. Our goal was not to impose a strict binary classification but to introduce these core conceptual dimensions. A single pruning method can indeed manifest elements of both paradigms.
>
> **(4)** We have readily shared the code in the supplementary material upon submission, and we are prepared to release it publicly via a GitHub repository upon acceptance, in order to preserve the double-blind review policy. Moreover, we have updated the code with the DWF implementation.
>
> ### Requested Changes
>
> **(1)** Thank you for the comment. CoNNect is primarily designed for structured pruning - we explicitly target the connectivity between functional modules within a network because it enables real computational speedup on standard hardware and aligns better with practical deployment pipelines. Unstructured pruning emerges as a natural special case, where each individual parameter is treated as a module. For example, in channel pruning, the scaling factor is typically applied per channel; by assigning one scaling factor per weight instead, CoNNect becomes compatible with unstructured pruning. In the paper, we adopt per-parameter notation mainly for simplicity and clarity of exposition, while the actual implementation and experiments focus on structured units. We hope this clarifies the intended scope of our method.

---

> > ### Author Response · Authors · 2025-07-23
> > **Response on Weaknesses and Requested Changes (2/2)**
> >
> > **(2)** We appreciate the reviewer’s question regarding Axiom 2. In our view, structured pruning refers to the systematic removal of channels or modules to enable practical speedups on standard hardware. However, dynamically adapting the surrounding layers to compensate for the removed modules is not an easy practice in most pruning pipelines. Moreover, when structured importance scores are computed independently for each module (e.g., channel), and global pruning is applied, it is possible for many or all removable modules to concentrate in the same layer - leading to collapse or severely degraded connectivity. Even in less extreme cases, as illustrated by our toy example, layer mismatches can occur when there is an imbalance in pruned inputs and outputs within the same layer, resulting in disconnected or invalid subnetworks, despite compatible input/output dimensions. Therefore, Axiom 2 remains essential for ensuring the preservation of meaningful end-to-end connectivity, especially under global structured pruning strategies.
> >
> > **(3)** We thank the reviewer for highlighting these redundancy elimination methods, which we had not previously considered. Upon review, we believe they can be accommodated within our existing categorization in the Related Work section. [1] and [2] employ regularization strategies during training to prepare the model for pruning, and thus we have added them under the soft pruning (regularization-based) category. [3] proposes a criterion for filter removal, applied post-training without affecting the training dynamics, and is therefore best classified as a stage-based (hard) pruning method. We have now incorporated these references into the revised manuscript accordingly.
> >
> > **(4)** Thank you for this question. When maximizing connectivity using the scaling factors introduced in Section 3.3.2, the regularizer will also operate on these normalized scaling factors associated with each channel. Similar to the arguments made on for CoNNect on the weight-level in Theorem 3.1, the normalization ensures that one must push certain scaling factors high and set others to 0 as to maximize the connectivity. In fact, one can show that the optimally connected network following Equation 4, including scaling factors for channel-level pruning, yields a NN using a single channel in each layer. If the reviewer finds it helpful, we can make this more precise in a revision.
> >
> > ### Minor Comments
> >
> > **(1)** We followed the reviewer’s suggestion and revised the abstract accordingly.
> >
> > **(2)** We thank the reviewer for pointing this out. In the revised manuscript, we now clarify that this refers to classical magnitude-based unstructured pruning approaches, where individual weights are pruned based on the L1 norm of weights.
> >
> > **(3)** We have added a clarification of $k$ denoting the layer index in the revision.
> >
> > **(4)** Indeed, $W_{i,j}$ refers to a single weight connecting neuron $i$ to neuron $j$. We take the absolute value of each weight because both positive and negative weights can carry significant influence on the network’s output, thereby contributing towards connectivity. In contrast, a weight value of zero means no signal is transmitted along that connection, and so there is no contribution to connectivity on that link. Therefore, the magnitude (i.e., absolute value) serves as a natural measure of signal strength or contribution to the overall (connectivity) computation.
> >
> > **(5)** Thank you for pointing this out. We have corrected the wording in the revised manuscript.
> >
> > **(6)** We agree with the reviewer and made a revision accordingly.
> >
> > **(7)** Thank you for pointing out this typo.
> >
> > **(8)** We thank the reviewer for the question. This statement refers specifically to the forward pass used to compute connectivity, not the actual model weights. During this connectivity computation, biases are set to zero to ensure that the signal flow depends solely on the weights. This does not imply that the network itself cannot use biases - bias terms remain part of the model during standard training and inference.
> >
> > **(9)** We agree the sentence was somewhat convoluted and have clarified the sentence in the revised manuscript: *Also, the regularizer does not cause input nodes to disconnect, as it does not depend on how many input nodes are connected to the second layer*.
> >
> > > References:
> > > [1] Kolb, Chris, Tobias Weber, Bernd Bischl, and David Rügamer. "Deep weight factorization: Sparse learning through the lens of artificial symmetries." ICLR 2025.
> > > [2] Sauron U-Net: Simple automated redundancy elimination in medical image segmentation via filter pruning. Pattern Recognition 2024.
> > > [3] Centripetal sgd for pruning very deep convolutional networks with complicated structure. CVPR 2019.
> > > [4] Filter pruning via geometric median for deep convolutional neural networks acceleration. CVPR 2019.

---

> > > ### Comment · Reviewer_s6Vh · 2025-07-28
> > > **Satisfied with the answer**
> > >
> > > ## Weaknesses
> > > (1) I think that this point became much clearer after the discussion and revision. My understanding is that, for every channel, you would end up having a single value that, as in the unstructured pruning case, you can leave/remove based on a threshold. However, in the third paragraph of Section 3.3.2, you define $\delta^{(k)}$ where $k$ refers to a layer. Wouldn't this $\delta$ be a different value for each channel to allow for filter pruning? Unless you would like to do structured pruning by removing entire layers.
> > >
> > > (2) I'm in favor of including this comparison in the paper.
> > >
> > > (3) I understand the author's view, and I personally do not like this way of categorizing previous work because of the fuzziness that this categorization introduces. According to this categorization, all methods from the first category also belong to the second, because, in practice, you need to eliminate the models' weights at some point (e.g., at the end of the training); otherwise, if you leave the weights as 0s, you are not really pruning, you are still doing operations, ultimately achieving no speed gains or reducing the models' size. Since this paper introduces a regularization, I would categorize previous pruning methods in "regularization-based" vs. "non-regularization based", and within the regularization-based you could talk about two trends: a) regularizers that lead to zero weights, and b) regularizers that increase to redundancy.
> > >
> > > I want to remark that this is the categorization of previous work that I would personally do in this particular paper because I genuinely think that it puts it in a context where it's easier to understand. I have no strong opinion on this matter, and I do not consider that this would be ground for rejection at all.
> > >
> > > (4) That's fantastic.
> > >
> > > ## Requested Changes
> > > I'm satisfied with the clarifications and changes made by the authors.

---

> > > > ### Author Response · Authors · 2025-07-31
> > > >
> > > > We thank the reviewer for the positive feedback. We hope to address the remaining concern below.
> > > >
> > > > **(1)** Your understanding is correct. We define $\delta^{(k)} \in \mathbb{R}^d$ to be a vector containing the scalar factors for the $d$ different channels in layer $k$. In this way, we indeed allow for filter pruning, as channel $i$, for $i = 1, \ldots, d$, is scaled by a factor $\delta_i^{(k)}$, which is independent of the scaling factors of other channels, such as $\delta_j^{(k)}$.
> > > >
> > > > **(2)** We will make sure to include these results in a future revision, to be released along with any other adjustments required by all reviewers.
> > > >
> > > > **(3)** We thank the reviewer for the thoughtful comment and for clarifying that this point is not a major concern. We also appreciate the alternative categorization suggested, and we will adopt the non-regularization-based terminology instead of stage-based in the revision.

---

### Decision · Action_Editor_Jipr · 2025-08-28

**Recommendation:** Accept as is

**Additional Comments:**

Reviewers highlighted the simplicity of the method and appreciated the quality of the writing. Some concerns regarding the clarity of the paper were raised during the initial submission, but reviewers acknowledged that these issues were resolved during the rebuttal phase.

In summary, I recommend acceptance of the paper.

**Audience:**

Yes

**Audience Explanation:**

Structural pruning is certainly a topic of interest for the machine learning community.

**Claims And Evidence:**

Yes

**Claims Explanation:**

The paper provides evidence that support the main claims of the paper that the proposed method preserves essential network structure and and improves upon established one-shot pruning methods.